# Molecular mechanisms underlying menthol binding and activation of TRPM8 ion channel

Lizhen Xu[1,10], Yalan Han[2,3,10], Xiaoying Chen[1], Aerziguli Aierken[1], Han Wen[4], Wenjun Zheng[4], Hongkun Wang[5,6], Xiancui Lu[2,3], Zhenye Zhao[1], Cheng Ma[7], Ping Liang [5,6], Wei Yang[8 ✉], Shilong Yang[9 ✉] & Fan Yang [1 ✉]

Menthol in mints elicits coolness sensation by selectively activating TRPM8 channel. Although structures of TRPM8 were determined in the apo and liganded states, the menthol-bounded state is unresolved. To understand how menthol activates the channel, we docked menthol to the channel and systematically validated our menthol binding models with thermodynamic mutant cycle analysis. We observed that menthol uses its hydroxyl group as a hand to specifically grab with R842, and its isopropyl group as legs to stand on I846 and L843. By imaging with fluorescent unnatural amino acid, we found that menthol binding induces wide-spread conformational rearrangements within the transmembrane domains. By Φ analysis based on single-channel recordings, we observed a temporal sequence of conformational changes in the S6 bundle crossing and the selectivity filter leading to channel activation. Therefore, our study suggested a 'grab and stand' mechanism of menthol binding and how menthol activates TRPM8 at the atomic level.

[1] Department of Biophysics, and Kidney Disease Center of the First Affiliated Hospital, Zhejiang University School of Medicine, 310058 Hangzhou, Zhejiang Province, China. [2] Key Laboratory of Animal Models and Human Disease Mechanisms of Chinese Academy of Sciences/Key Laboratory of Bioactive Peptides of Yunnan Province, Kunming Institute of Zoology, 650223 Kunming, Yunnan, China. [3] University of Chinese Academy of Sciences, Beijing, China. [4] Department of Physics, State University of New York at Buffalo, Buffalo, NY, USA. [5] Key Laboratory of Combined Multi-organ Transplantation, Ministry of Public Health, The First Affiliated Hospital, Zhejiang University School of Medicine, Zhejiang, China. [6] Institute of Translational Medicine, Zhejiang University, Zhejiang, China. [7] Protein facility, School of Medicine, Zhejiang University, Zhejiang, China. [8] Department of Biophysics, and Department of Neurosurgery of the First Affiliated Hospital, Zhejiang University School of Medicine, 310058 Hangzhou, Zhejiang Province, China. [9] College of Wildlife and Protected Area, Northeast Forestry University, 150040 Harbin, China. [10] These authors contributed equally: Lizhen Xu, Yalan Han. ✉email: yangwei@zju.edu.cn; syang2020@nefu.edu.cn; fanyanga@zju.edu.cn

People enjoy the pleasant freshness and coolness sensations of mints, which are largely mediated by the organic compound menthol. Therefore, menthol has been widely used as an analgesic to relieve acute, inflammatory, and neuropathic pain[1,2], as an antipruritic to reduce itching[3], and as an additive to soothe minor irritations in mouth or throat. Mechanistic understanding of menthol began to emerge after the cloning its receptor, the transient receptor potential melastatin 8 (TRPM8) ion channel[4,5].

TRPM8 channel is a polymodal receptor activated by a plethora of stimuli such as cold, membrane depolarization, and chemical ligands[6]. It is also critically involved in many pathological processes such as cold hyperplasia, prostate cancer, and migraine[7], making this channel a promising drug target[8]. However, though the inhibitors of TRPM8 like PF-05105679 effectively suppress cold-related pain[9], they also altered temperature sensation in patients, which limited their progress in clinical trials[8]. Therefore, understanding at the atomic level how menthol, the most classic agonist of TRPM8, binds and activates this channel will not only shed new light on its ligand-gating mechanism but also help developing modality-specific therapeutics targeting this channel.

To probe how menthol binds to TRPM8, previous studies have shown that mutations of residues within the transmembrane domains and the TRP domain such as Y745, R842, Y1005, and L1009 (all residue numbering is based on mouse TRPM8) largely disrupt menthol activation[10,11]. Radio-active menthol-binding assay further showed that this molecule binds to the vicinity of S2 transmembrane domain[11]. In the recent high-resolution cryo-EM structures of an avian TRPM8 homolog channel[12,13], WS-12 (a chemical analog of menthol) also binds within the cavity formed by S1–S4 segments. Therefore, all these studies have clearly suggested the cavity near Y745 and R842 as the binding pocket for menthol. However, though TRPM8 structures are available[12–14] in the apo (PDB ID: 6BPQ and 6O6A), WS-12 bound (PDB ID: 6NR2), icilin bound (PDB ID: 6NR3 and 6NR4), antagonist bound (PDB ID: 6O6R and 6O72), and calcium-bound (PDB ID: 6O77) states, due to the lack of a TRPM8 structure in menthol-bound state, both the binding configuration of menthol within this pocket and how menthol binding further triggers channel activation remain virtually unknown.

To answer these two questions, we employ an experimental strategy where computational and functional analyses were combined, which we have successfully employed to study the capsaicin activation of TRPV1 channel[15–17]. We first dock the menthol molecule to its putative binding pocket to probe its binding configurations and potential interactions with the channel protein. We then test the predicted interactions with patch-clamp recording and thermodynamic mutant cycle analysis. By altering the chemical structure of menthol with its analogs and introducing point mutations in TRPM8, we calculate the coupling energy between specific atoms in menthol and residues in TRPM8. From these experiments we determine the binding configuration of menthol. We further systematically introduced unnatural amino acid 3-(6-acetylnaphthalen-2-ylamino)-2-aminopropanoic acid (ANAP) throughout the transmembrane domains to probe the conformational changes upon menthol binding. By further performing Φ analysis based on single-channel recordings, we observe a temporal sequence of conformational changes in the S6 bundle crossing and the selectivity filter leading to channel activation.

## Results

**Potential menthol–TRPM8 interactions revealed by docking.** Among the eight stereoisomers, (−)-menthol is the naturally

existing form so that we examined its interaction with TRPM8 channel throughout this study. We named the hydroxyl group in menthol as the hand and its isopropyl group as the legs (Fig. 1a). In our patch-clamp recordings, menthol activated wild-type TRPM8 with an $EC_{50}$ 185.4 ± 69.4 μM (Hill coefficient: 1.74 ± 0.06, $n = 5$), but the previously reported Y745H mutation virtually abolished menthol activation[10] (Fig. 1a, b). We further observed that menthol served as a partial agonist of TRPM8 based on noise analysis of macroscopic current (Fig. 1c), which was also confirmed in single-channel recordings (Supplementary Fig. 1a, b). At near saturating concentration (500 μM), the maximum open probability ($P_{o\ max}$) of menthol is clearly less than unity (0.81 ± 0.03, $n = 5$). The partial agonist nature of menthol allowed us to accurately calculate its binding affinity ($K_d$) from concentration–response curves later.

To understand how menthol binds to TRPM8, we first examined the proposed binding pocket formed by the S1–S4 transmembrane segments (Fig. 1d). The topology of a single subunit of the TRPM8 homotetramer was illustrated in Supplementary Fig. 1c. According to the cryo-EM structure of TRPM8 (PDB ID: 6BPQ) in the apo state, residues critically affecting menthol activation (R842 and Y745) are tightly packed, which are further stabilized by electrostatic interaction with D802 as the distance between the nitrogen atom on the sidechain of R842 and the oxygen atom on the sidechain of D802 is only 2.5 Å (Fig. 1d, zoomed in)[12]. We employed the ConSurf server to analyze the sequence conservation in TRPM8 channel[18]. We observed that residues within the S1–S4 domain like R842, D802, Y745, L843, and I846 are well conserved in evolution, suggesting critical functional roles of these residues (Supplementary Fig. 2a, b). These results are in full consistency with structural studies of TRPM8 channel[12,13,19], so we believe that the ligand-binding pocket is formed within the S1–S4 domain.

We docked the menthol molecule into this pocket in the WS-12 and PI(4,5)P2 bound state (PDB ID: 6NR2) with the RosettaMembrane energy function in the RosettaLigand application[20–23]. All the top 10 models with lowest binding energy (Supplementary Fig. 3a) formed one cluster, where they were well converged (Supplementary Fig. 3b). The funnel-shaped binding energy profile indicated reliable docking (Supplementary Fig. 3c). Menthol was wedged in between Y745 and R842 with its isopropyl legs points downward, while its hydroxyl hand likely formed a hydrogen bond with the sidechain of R842 (Fig. 1e). Such a binding configuration of menthol would disrupt the packed conformations of Y745, R842, and D802 in the apo state, initiating conformational rearrangements from the binding pocket.

Moreover, we performed extra docking studies to test whether our docking method can recapture the native conformation of TRPM8 agonists and antagonists revealed in cryo-EM structures. We observed that for the agonist WS-12 and antagonists AMTB and TC-l 2014, our top models with largest binding energy converged well to the ligand-binding configuration observed in cryo-EM structures with RMSD values less than 2 Å (Supplementary Fig. 4a, c, d). For icilin, our docking recaptured the overall binding orientation of the molecule with deviations in binding location (Supplementary Fig. 4b). These results demonstrated that our docking protocol and the software can capture a generally correct binding configuration of the TRPM8 ligands.

To unveil the detailed interactions between menthol and TRPM8 channel, based on the top 10 docking models we further decomposed the binding energy from docking models to reveal spatial distribution of each energy components. We found that the binding was mainly determined by hydrogen bonds and van der Waals (VWD) interactions (Fig. 1f–h). While the potential VWD interactions were widely distributed within the binding

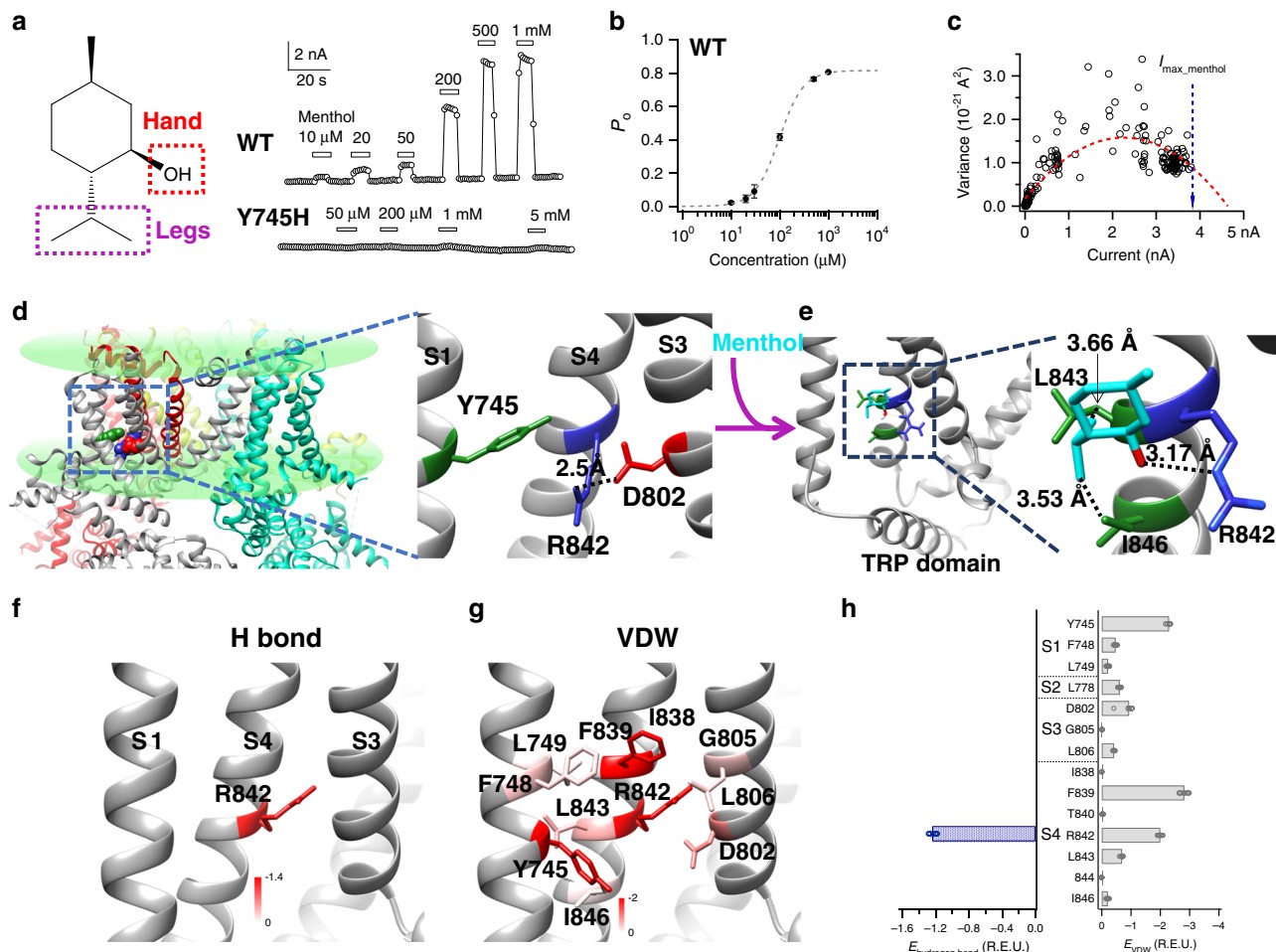

**Fig. 1 Potential menthol–TRPM8 interactions revealed by docking. a** The chemical structure of (−)-menthol, where its hydroxyl and isopropyl moieties are named as hand and legs, respectively. Menthol activated TRPM8 channel in a concentration-dependent manner in whole-cell patch-clamp recordings. **b** The concentration–response curve of menthol activation measured from whole-cell patch-clamp recordings ($n = 5$). **c** The maximum open probability ($P_{o\ max}$) was determined from noise analysis of menthol-induced TRPM8 current. The measured maximum current ($I_{max\_menthol}$) was normalized to the predicted maximum current to derive $P_{o\ max}$. **d** The putative menthol-binding pocket located within the transmembrane domains of TRPM8 channel as revealed by cryo-EM. The zoom-in view of the binding pocket illustrated that residues known to be critical for menthol activation are tightly packed in the apo state (PDB ID: 6BPQ). **e** Docking of menthol into the binding pocket in the WS-12-bound activated state (PDB ID: 6NR2) lead to disruption of residue packing. The hydroxyl hand of menthol is predicted to form a hydrogen bond with the sidechain of R842 (dashed line in red). **f–h** Breakdown of the menthol-binding energy (**h**). The per-residue energy contributed by hydrogen bonding (**f**) and VDW interactions (**g**) was mapped onto the structure of TRPM8, respectively. The redder in color scale indicates larger energy value in REU (Rosetta energy unit). All statistical data are given as mean ± s.e.m. Source data are provided as a Source Data file.

pocket, we found that residue R842 may form a hydrogen bond with menthol. We then performed patch-clamp recordings to functionally verify these predicted menthol-channel interactions.

**A specific interaction between hydroxyl hand and the channel.** If the hydroxyl hand of menthol interacts with TRPM8 (Fig. 2a), we expected that *p*-menthane, an analog of menthol but lacking the hydroxyl moiety (Fig. 2b), would show much reduced agonist effects. Indeed, in patch-clamp recordings we observed that TRPM8 channels were barely activated by *p*-menthane even up to 10 mM (Fig. 2c), which clearly demonstrated that the hydroxyl hand is required for robust activation of TRPM8.

To further elucidate whether the hydroxyl hand specifically interacts with R842 on TRPM8 as predicted by the docking, we performed thermodynamic mutant cycle analysis with patch-clamp recordings. Briefly, if the hydroxyl hand of menthol specifically interacts with a residue on TRPM8, either changing this group or mutating the residue on channel should show

nonadditive effects on binding affinity. Previous studies including our own have shown that if the measured coupling energy is larger than 1.5 $kT$ (or 0.89 kcal/mol at 24 °C)[15,24,25], a specific interaction can be reliably assumed. To perturb the hydroxyl group of menthol, we used a menthol analog named menthone, where the hydroxyl group is replaced by a carbonyl group (Fig. 2d). By using either menthol or menthone, we measured their concentration–response curves on either wild-type TRPM8 channel or the mutant channel R842K (Fig. 2d). From these curves we observed that menthone exhibited a largely increased EC50 (813.4 ± 84.3 μM, $n = 3$) while the maximum open probability was also decreased to 0.48 ± 0.05 ($n = 3$). We further gauged the energetic effects of shifting in concentration–response curves with a general ligand-gating scheme (Fig. 2e), where the ligand binding (represented by $K_d$) and subsequent conformational changes leading to channel opening (represented by $L$) were separately quantified. As menthol is a partial agonist for TRPM8 with a $P_{o\ max}$ smaller than unity (Fig. 1c, Supplementary Fig. 1), we can directly and accurately calculate $K_d$ and $L$ from

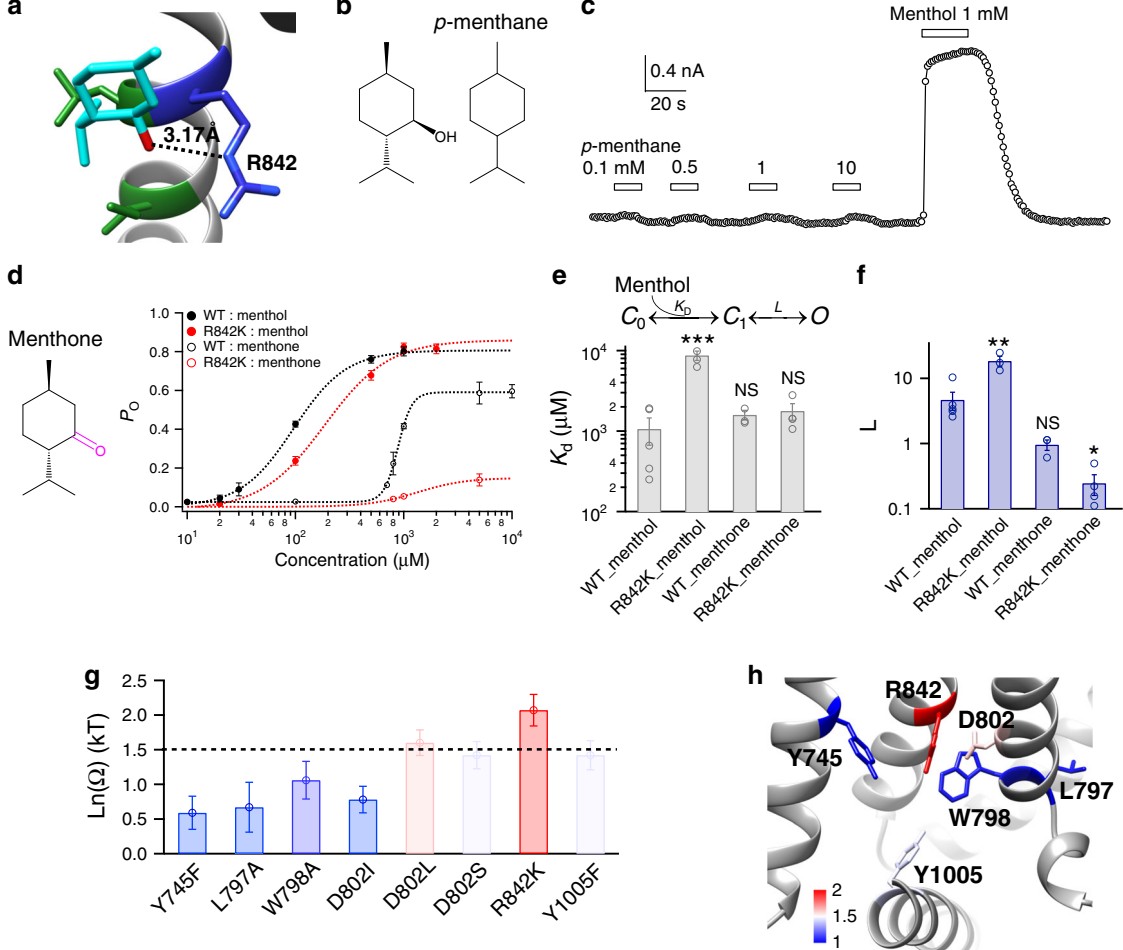

**Fig. 2 A specific interaction between menthol hydroxyl hand and the channel. a** Docking predicted that there is a hydrogen bond between the hydroxyl hand of menthol and the sidechain of R842 (dashed line in black). **b** Chemical structures of menthol and *p*-menthane. **c** Representative whole-cell patch-clamp recording showed that *p*-menthane lacking the hydroxyl hand up to 10 mM cannot activate TRPM8 channel. **d** Chemical structure of menthone, which was used in thermodynamic mutant cycle analysis. The concentration–response curves of wild type and mutant such as R842 with either menthol or menthone were measured with whole-cell patch-clamp recordings ($n = 3$–5). **e, f** The general gating scheme where the ligand binding is represented by $K_d$ and the equilibrium constant between the closed and open states upon ligand binding is $L$. For wild type and R842 channel, $K_d$ and $L$ values were calculated from the concentration–response curves in **d** (two-sided *t*-test, *$p < 0.05$; **$p < 0.01$; ***$p < 0.001$; NS, not statistically significant). **g** Summary of coupling energy measurements. Coupling energy value was calculated from the $K_d$ values. Mutants showing a coupling energy larger than 1.5 kT (dashed line) were colored in red. Those with less energy were colored in different shades of blue. At least four independent trials were performed for each chemical at each concentration. **h** Spatial distribution of coupling energy values within the menthol-binding pocket. Color scheme is the same as in **g**. Color scale is in the unit of kT. All statistical data are given as mean ± s.e.m. Source data are provided as a Source Data file.

experimentally measured $EC_{50}$ and $P_{o\ max}$, because $EC_{50} = K_d/(1 + L)$ and $P_{o\ max} = L/(1 + L)$ as previously described[15]. The $K_d$ between menthol and R842K was increased, while menthone showed similar $K_d$ with both wild type and the mutant (Fig. 2e). This is not surprising because the oxygen atom in menthone reserves the capability to form a hydrogen bond with either wild type or mutant channels. Conversely, removing polar atoms from either menthol in our study by using methane or from the residue R842 by alanine substitution as reported before[11] lead to the elimination of ligand activation (Fig. 2c), preventing us from estimating changes in energy associated with breaking of this predicted hydrogen bond. Nevertheless, we later used the calculated $K_d$ values to measure the coupling energy between the menthol hydroxyl hand and residues on TRPM8.

We calculated coupling energy values for a series of mutants with either menthol or menthone (Fig. 2g). We observed that the mutant R842K showed a coupling energy much larger than the

1.5 kT threshold, indicating that there is a specific interaction between this residue and the hydroxyl hand of menthol as predicted by our docking experiments (Fig. 2a). Moreover, the D802 residue also exhibited a coupling energy value larger than the threshold. We believe that this is more likely caused by the interaction between D802 and R842 in the apo state as revealed by the cryo-EM structure of TRPM8 (Fig. 1f)[12], so that D802 interacts with the hydroxyl hand of menthol indirectly through R842. In contrast, Y745 and other residues within the binding pocket showed coupling energy values less than the threshold (Fig. 2g, f). These results first confirmed that the cavity formed between S1 and S4 is indeed the binding pocket for menthol, as residues here show specific interactions with the hydroxyl hand of menthol. Furthermore, as the $L$ values for menthone on the wild type and R842K channels were reduced (Fig. 2f), the specific interaction between R842 and menthol contributes to both ligand binding and gating transition to activate TRPM8.

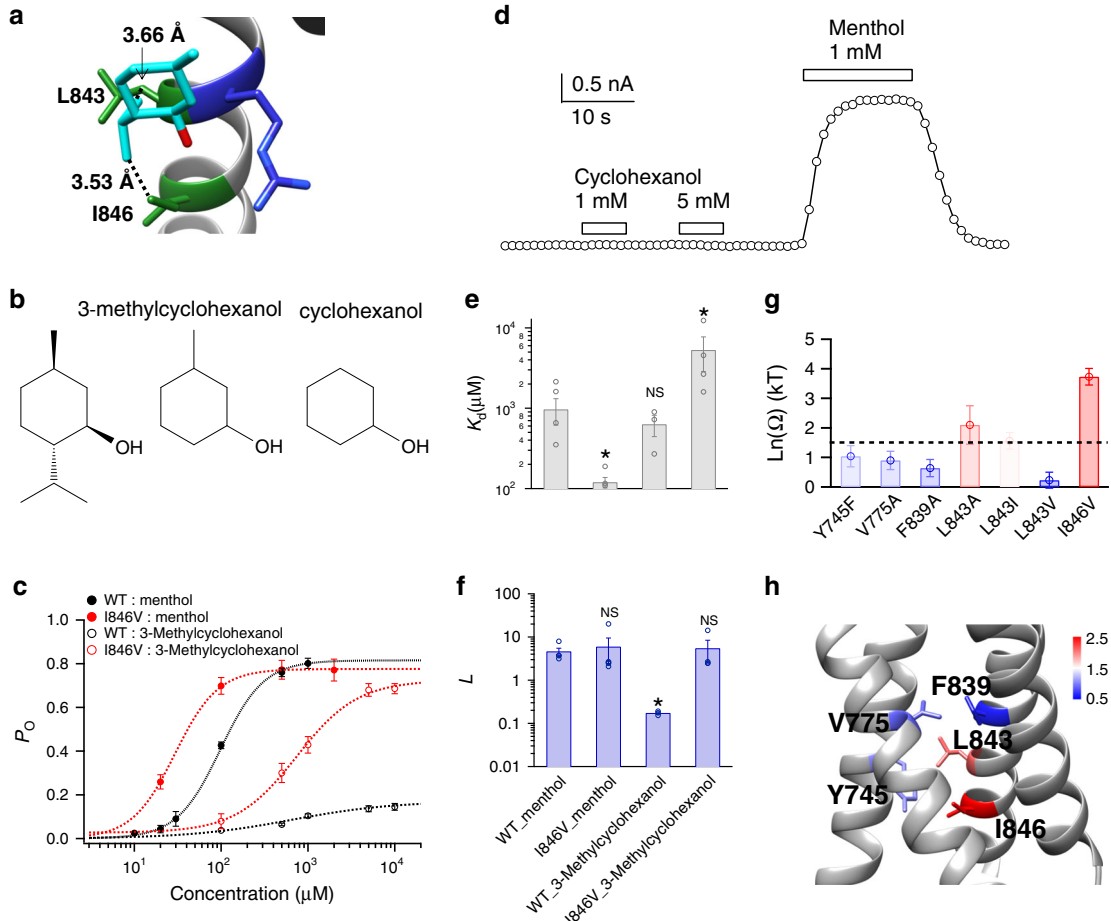

**Fig. 3 Specific interactions between menthol isopropyl legs and the channel probed with 3-methylcyclohexanol. a** Docking predicted that the isopropyl legs of menthol stand on hydrophobic residues through VDW interactions (dashed lines in black: C9 atom of menthol to CD1 atom of I846: 3.53 Å; C8 atom of menthol to CB atom of L843: 3.66 Å). **b** Chemical structures of menthol, 3-methylcyclohexanol lacking the isopropyl legs, and cyclohexanol where both the isopropyl and methyl moieties are missing compared to menthol. **c** The concentration–response curves of wild type and mutant such as I846V with either menthol or 3-methylcyclohexanol were measured with whole-cell patch-clamp recordings ($n = 3$–5). **d** Representative whole-cell patch-clamp recording showed that cyclohexanol up to 5 mM cannot activate TRPM8 channel. **e, f** For wild type and I846V channel, $K_d$ and $L$ values were calculated from the concentration–response curves in **c** (two-sided $t$-test, *$p < 0.05$; NS not statistically significant). **g** Summary of coupling energy measurements. Coupling energy value was calculated from the $K_d$ values. Mutants showing a coupling energy larger than 1.5 kT (dashed line) were colored in red. Those with less energy were colored in different shades of blue. At least four independent trials were performed for each chemical at each concentration. **h** Spatial distribution of coupling energy values within the menthol-binding pocket. Color scheme is the same as in **g**. Color scale is in the unit of kT. All statistical data are given as mean ± s.e.m. Source data are provided as a Source Data file.

**Specific interactions between isopropyl legs and the channel.** Besides the specific interaction between the hydroxyl hand of menthol and the channel protein, our docking results also predicted wildly distributed VDW interactions (Figs. 1h and 3a). To test these interactions, we aimed to remove the non-polar moieties in menthol as they most likely contribute to ligand binding with VDW interactions. We employed two chemical analogs of menthol, where the isopropyl moiety is removed (Fig. 3b, 3-methylcyclohexanol) or both the isopropyl and methyl groups are removed (Fig. 3b, cyclohexanol). In patch-clamp recordings, we observed that 3-methylcyclohexanol activated TRPM8 wild type and mutant channels with much reduced $P_{o\ max}$ (Fig. 3c), while cyclohexanol up to 5 mM failed to open TRPM8 (Fig. 3d). These observations illustrated that the VDW interactions by the non-polar groups in menthol is essential for channel activation.

To reveal the detailed interactions between menthol isopropyl legs and the channel, we performed thermodynamic mutant cycle analysis with 3-methylcyclohexanol. We first calculated the $K_d$ and $L$ values (Fig. 3e, f) from concentration–response curves

based on the general ligand-gating scheme (Fig. 2e). We further calculate the coupling energy between the isopropyl group and the channel based on the $K_d$ values. We observed coupling energy values larger than the 1.5 kT threshold at L843 and I846 (Fig. 3g, h). In contrast, V775 outside the binding pocket exhibited a small coupling energy with the isopropyl legs. Y745 and F839 located within the binding pocket, but as they were apart from the isopropyl legs, they exhibited smaller coupling energy as compared to those of I846 and L843 (Fig. 3g, h). These results suggested that the isopropyl legs point downward as predicted in the docking experiments (Fig. 3a), where they stand on several residues including L843 and I846 to facilitate channel activation.

Removing the isopropyl legs from menthol may affect its binding configuration and compromise the thermodynamic mutant cycle analysis. To further test the interaction between the isopropyl group of menthol and TRPM8 channel, we employed isopulegol, a menthol analog where the isopropyl group is not removed but replaced with a methylethenyl group (Fig. 4a, dashed box in red). We performed thermodynamic

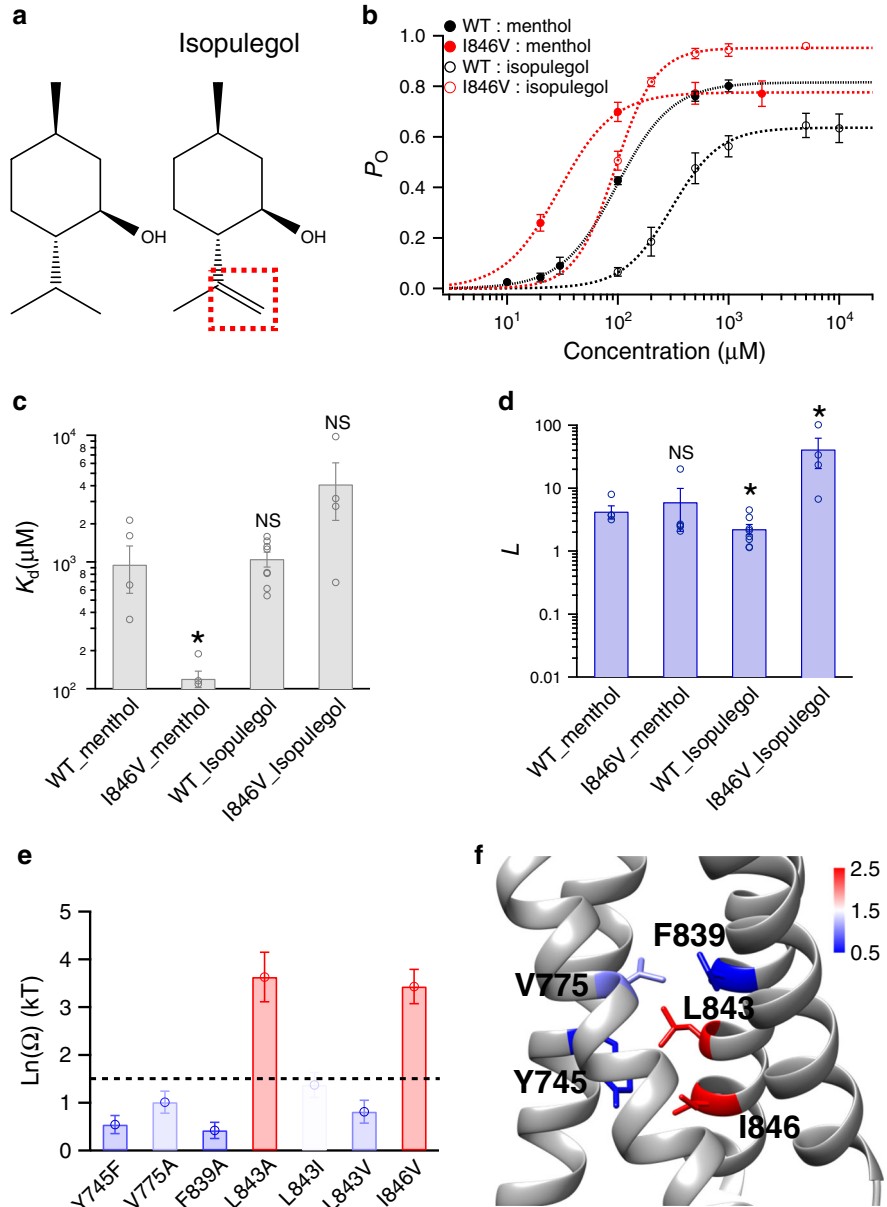

**Fig. 4 Specific interactions between menthol isopropyl legs and the channel probed with isopulegol. a** Chemical structures of menthol and isopulegol. The isopropyl group in menthol is replaced by the methylethenyl group in isopulegol (dashed box in red). **b** The concentration–response curves of wild type and mutant such as I846V with either menthol or isopulegol were measured with whole-cell patch-clamp recordings ($n = 3–5$). **c, d** For wild type and I846V channel, $K_d$ and $L$ values were calculated from the concentration–response curves in **c** (two-sided $t$-test, *$p < 0.05$; NS not statistically significant). **e** Summary of coupling energy measurements. Coupling energy value was calculated from the $K_d$ values. Mutants showing a coupling energy larger than 1.5 kT (dashed line) were colored in red. Those with less energy were colored in different shades of blue. At least four independent trials were performed for each chemical at each concentration. **f** Spatial distribution of coupling energy values within the menthol-binding pocket. Color scheme is the same as in **e**. Color scale is in the unit of kT. **g** Structural fluctuation in the menthol-bound S1–S4 domain during 378 ns molecular dynamic simulation. Root-mean-square deviation (RMSD) of protein structures was calculated as compared to our menthol docking model. **h** Ensemble plot of the menthol molecule (dashed box in yellow) with snapshots of the simulation from the beginning to the end (red and blue, respectively). A snapshot was saved every 10 ns during the simulation. All statistical data are given as mean ± s.e.m. Source data are provided as a Source Data file.

mutant cycle analysis with isopulegol (Fig. 4 b–d) and observed large coupling energy values at L843 and I846 sites (Fig. 4e). A coupling energy larger than 1.5 kT predicts that the distance between interacting parties is less than 4 Å[24]. Indeed, our docking results showed the carbon atoms in the isopropyl group of menthol are about 3.53 and 3.66 Å from the sidechains of I846 and L843 (C9 atom of menthol to CD1 atom of I846: 3.53 Å; C8 atom of menthol to CB atom of L843: 3.66 Å), respectively (Fig. 1e). The spatial distribution of coupling energy measured

with isopulegol was very similar to that measured with 3-methylcyclohexanol (Figs. 4f and 3h, respectively), so we believe that our thermodynamic mutant cycle analysis with menthol and its analogs is robust to probe ligand–protein interactions.

**Menthol-induced conformational changes revealed by ANAP.** Our thermodynamic mutant cycle analysis and computational studies above have revealed that menthol employs a "grab and stand" mechanism to bind with TRPM8 through its hydroxyl

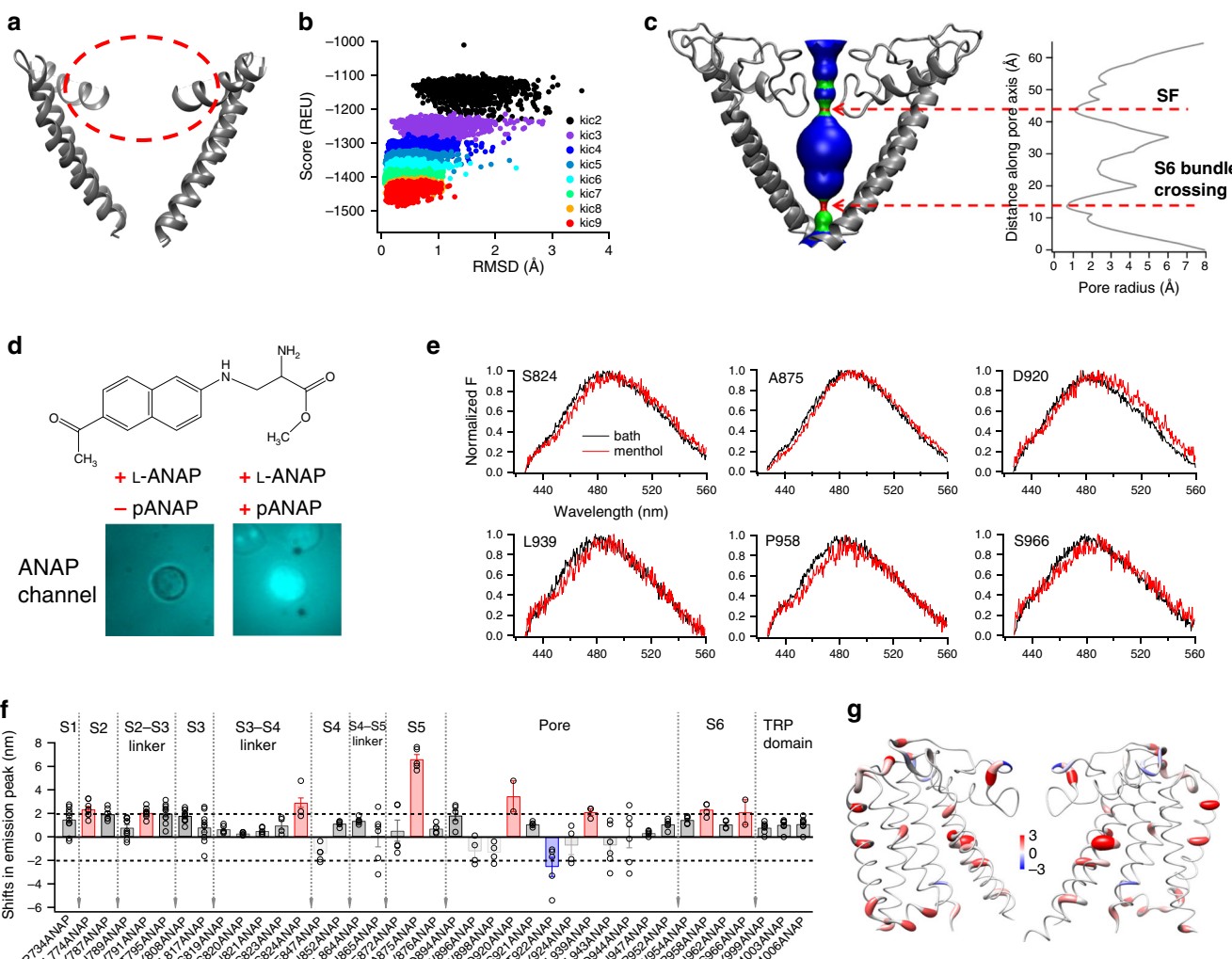

**Fig. 5 Menthol-induced conformational rearrangements revealed by ANAP imaging. a** The selectivity filer of TRPM8 was missing in its cryo-EM structure (PDB ID: 6BPQ). It is expected to locate within the dashed circle in red. **b** The models of selectivity filter after nine rounds of KIC loop modeling exhibited a funnel-shaped distribution of total energy calculated by Rosetta (REU, Rosetta energy unit). **c** Distribution of pore radii in TRPM8 with the selectivity filter modeled. The regions in red are too narrow to allow a water molecule to pass. The regions in green or blue are sites allowing single or multiple water molecules and ions to pass, respectively. Pore radii were calculated by the HOLE program. SF selectivity filter. **d** Chemical structure of ANAP and representative images of negative control cells where the pANAP vector was not added, and positive cells expressing ANAP-incorporated TRPM8. Pseudocolors for ANAP were used. Scale bar: 10 μm. **e** Representative emission spectra (black, bath solution; red, solution containing 0.5 mM menthol) of ANAP-incorporated TRPM8 mutants. **f** Summary of shifts in ANAP emission peak at different incorporation sites (for each ANAP-incorporation site, $n =$ 4–6). Right and left shifts larger than 2 nm were colored in red and blue, respectively. **g** Shifts in ANAP emission peak mapped onto the TRPM8 model with a worm representation, where the worm radius was proportional to the amplitude of shifts. Right and left shifts were colored in red and blue, respectively. Color scale is in the unit of nanometer. All statistical data are given as mean ± s.e.m. Source data are provided as a Source Data file.

hand and isopropyl legs. We next investigated upon binding, how menthol induces conformational rearrangements.

To understand the conformational transitions in TRPM8, we first aimed to reveal a model of TRPM8 channel in the closed state. Though cryo-EM studies have determined several structures of TRPM8 channel[12–14], the selectivity filter always remained missing in the apo and agonist- and antagonist-bound states (Fig. 5a). This domain was only observed in cryo-EM structure in the calcium-bound desensitized state[14] (PDB ID: 6O77), so we needed to first determine the conformation of this domain in both closed and menthol-activated open states. Previously we have computationally modeled the missing selectivity filter region with multiple rounds of de novo loop modeling procedures in the Rosetta suite (Fig. 5b)[26]. In our model, the S6 bundle crossing was in its original conformation as determined in cryo-EM studies[12,13] since our modeling procedure was limited to the

selectivity filter region. We analyzed our model with selectivity filter complemented and observed that the pore radii there and near the S6 bundle crossing were too small to allow ions and water molecules to pass (Fig. 5c, regions in red). Despite the small pore radius at the selectivity filter, in other TRP channels like TRPV1 to TRPV3 this domain is not a gate for ion permeation[27]. However, the S6 bundle crossing is a well-established gate in TRP channels[12,28,29]. Therefore, we believed that our model most likely represents the TRPM8 channel in a closed state (Fig. 5c).

To probe menthol-induced conformational changes, we employed the fluorescence unnatural amino acid ANAP[30], which we have successfully used to reveal conformational changed induced by capsaicin in TRPV1 channel[17]. As fluorescence emission peak of ANAP shifts to a longer wavelength in more hydrophilic environment[30], we measured the shift in ANAP emission peak to report conformational changes. Fluorescence

properties of ANAP may also be affected by other factors such as temperature and pH[31], so we performed our ANAP experiments always at room temperature (22 ± 1 °C). To control for changes in solution pH, we added 3 mM HEPES in ANAP imaging solution to buffer the pH at 7.2. We genetically integrated ANAP into a specific site on TRPM8 with special tRNA and synthetase encoded by pANAP vector (Fig. 5d). We incorporated ANAP at 73 sites throughout the transmembrane domains (S1–S6) and the TRP domain, but ANAP incorporation was tolerated only at 39 sites where a mutant channel with ANAP was still activated by menthol (Supplementary Table 1). Among these functional ANAP-incorporated channels, we observed eight redshifted and one blueshifted mutants with menthol-induced changes in ANAP emission peak larger than 2 nm (Fig. 5e, f), which was the detection limit in our imaging system. For instance, at A875 site the ANAP emission peak was redshifted by 6.6 ± 0.4 nm ($n = 5$). We believed that there were conformational changes near these nine sites, so we mapped them onto the model of TRPM8 in the closed state (Fig. 5f, g). L774, V791, and S824 locates in the S2, S2–S3 linker, and S3–S4 linker near the extracellular end of S4, respectively, whose redshift in ANAP emission reflects conformational changes in the ligand-binding pocket induced by menthol binding to R842. Furthermore, A875, D920, T922, L939, P958, and S966 locate within the pore region formed between S5 and S6, so their shifts in ANAP emission suggested widely distributed conformational rearrangements in the pore region.

**Conformational dynamics in menthol activation.** To further probe the dynamic conformational changes induced by menthol to activate TRPM8, we performed the rate-equilibrium linear free-energy relationships analysis (Φ analysis) with single-channel recordings. Previously we have applied this analysis to reveal the conformational wave elicited by capsaicin in the TRPV1 channel[17]. We first introduced point mutations to residues near the ligand-binding pocket, the selectivity filter, and the S6 bundle crossing of the TRPM8 channel, as these mutations brought asymmetrical energetic effects on the equilibrium between the closed and open states (Fig. 6a). We quantified such energetic effects with single-channel recordings (Fig. 6b). By measuring opening and closing rates from single-channel recordings (Fig. 6c–f), we calculated a Φ value ranging from zero to one for one specific residue. We observed that residues near the selectivity filter showed smaller Φ values (sites 910 and 916: 0.63 and 0.42, respectively) than residues near the ligand-binding pocket or the S6 bundle crossing gate (sites 840 and 972: 0.73 and 0.71, respectively) (Fig. 6g–j). There are multiple ways to interpret measured Φ values, for instance, when there were more than one parallel transition pathways between closed and open states, the Φ value for a specific position reflects the probability that a particular activation pathway is taken. In a simplified sense, the Φ value reflects the relative position of the transition state along a single reaction pathway: if a residue moves earlier during menthol activation, then its Φ value would be larger; a residue moving later has a smaller Φ value. Therefore, our Φ analysis most likely suggested that upon menthol binding, the ligand-binding pocket and the S6 bundle crossing gate move first, while the selectivity filter region moves later to fully open the channel (Fig. 6k).

Moreover, we computationally built a potential menthol-induced open state model of the channel by integrating ANAP fluorescence information as experimentally derived constraints to filter out inconsistent computational models (Supplementary Fig. 7, see Methods for details). We acknowledge that due to the flexibility of selectivity filter in TRPM8 revealed by the cryo-EM structures[12–14] and uncertainties brought into our ANAP imaging experiments by factors such as lipid composition of cell

membrane and the state of the channel-expressing HEK293 cells, our model is not an unambiguous atomic model of menthol-bound open state, but it only represents one energetically stable state of the channel that is compatible with our ANAP imaging experiments (Fig. 5f, Supplementary Fig. 7). Nevertheless, we could gain insights into the menthol gating mechanism by comparing our potential models of TRPM8 channel in the closed and open states. Consistent with our ANAP imaging results (Fig. 5f, g), menthol caused structural changes throughout the transmembrane domains. Binding of menthol into the cavity formed by S1 to S4 led to disruption of interactions between Y754 in S1, D802 in S3, and R842 in S4 in the closed state (Fig. 1d, e). We further hypothesized that since R842 serves as part of the total gating charge in channel activation by depolarization[11], hyperpolarization would stabilize R842 in its resting state, which may prevent conformational changes induced by menthol bounded to this residue. Consistent with this hypothesis, we observed that deep hyperpolarization beyond −200 mV closed the channel even in the presence of near saturating concentration of menthol (Supplementary Fig. 8a). In contrast, capsaicin activation of TRPV1 was not antagonized by deep hyperpolarization (Supplementary Fig. 8b), because capsaicin is known to activate the channel without involving voltage-sensing apparatus, though these two stimuli are allosterically coupled[32,33]. Menthol binding may further lead to widening of the S6 bundle crossing (Supplementary Fig. 7e).

To corroborate our observations of conformational dynamics in Φ analysis, we resorted to the computational technique interpolated elastic network modeling (iENM)[34], where amino acids in a protein are represented as spheres to efficiently calculate the transition pathway between two states. iENM has successfully predicted temporal sequence of events in ligand activation of TRPV1 channel[15,17] and pentameric ligand-gated ion channels[34,35]. From the iENM calculation, we observed that the residues in the S2–S3 linker, S3, and S4 forming the menthol-binding pocket showed largest $f_{progress}$ values, indicating they moved earliest in time. Upon menthol binding, residues near the S6 bundle crossing moved earlier than those around the selectivity filter (Fig. 6l). We also noticed that certain parts in the S1–S4 domain like the upper S2 showed movements later than the S6 bundle crossing and the selectivity filter (Fig. 6l), such late motion may reflect the continuing movements of channel protein to accommodate conformational rearrangements in the selectivity filter and S6 bundle crossing upon menthol binding. Our calculated $f_{progress}$ values from iENM matched well with the Φ values measured from single-channel recordings (Fig. 6m). ConSurf analysis revealed that residues around the selectivity filer like P916 are evolutionarily conserved (Supplementary Fig. 2c, d). This is consistent with our functional studies as P916 locates at the entrance of selectivity filter and moved later in time during menthol activation (Fig. 6k). Therefore, we believed that similar to the ligand activation of TRPV1 channel[17] and nicotinic acetylcholine receptors[35,36], menthol initiates a conformational wave starting from its binding pocket, which propagates first through the S4–S5 linker towards the S6 bundle crossing and then to the selectivity filter to activate TRPM8 channel (Fig. 6n).

## Discussion
With docking and patch-clamp recordings, we have established that menthol binds to the cavity formed between S1 and S4 of TRPM8 channel with a "grab and stand" mechanism, where its hydroxyl group works as a hand to specifically grab with R842 likely through a hydrogen bond, while its isopropyl legs stand on residues on S4 through VDW interactions. Menthol binding induced widespread conformational rearrangements as reported

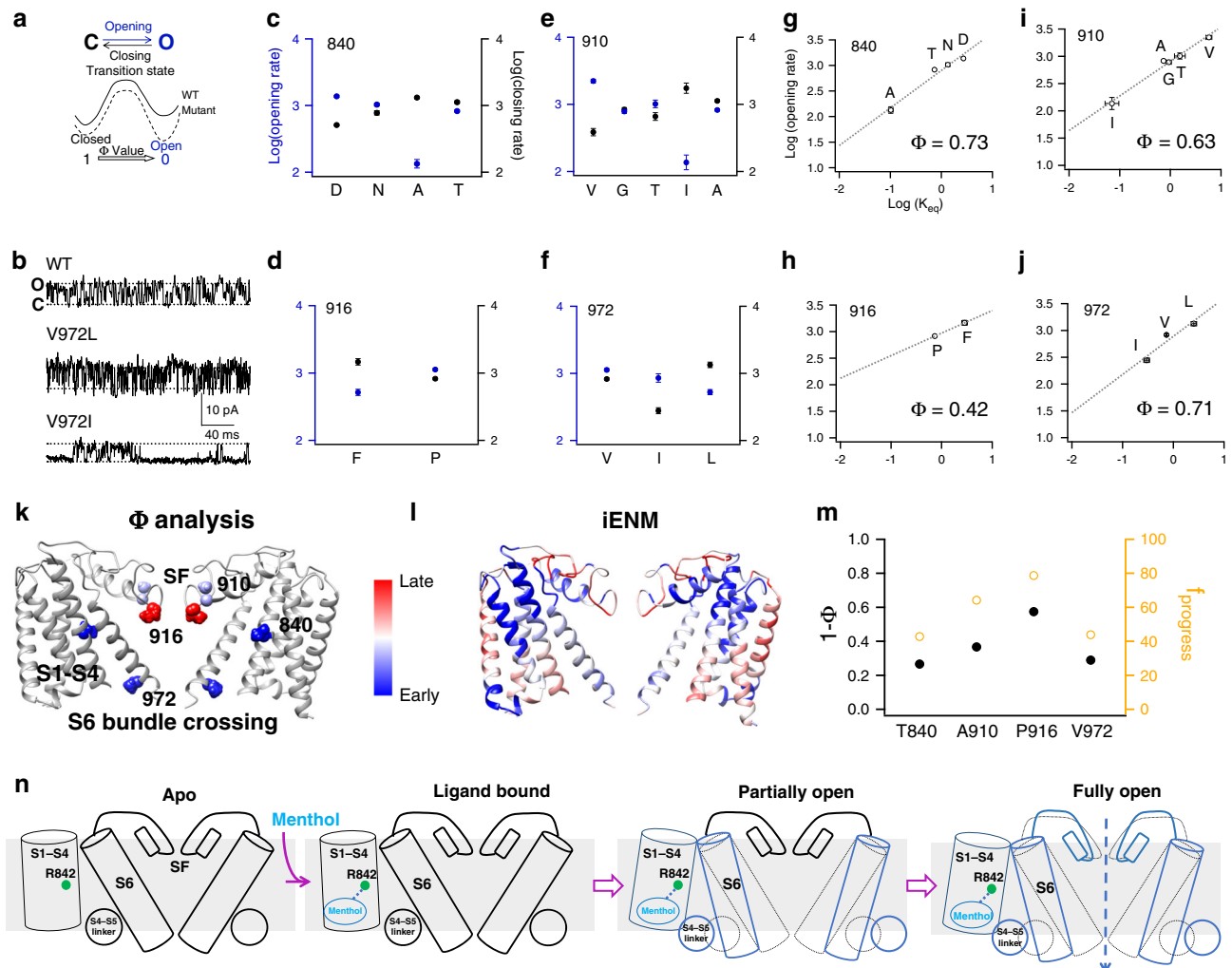

**Fig. 6 Φ analysis and temporal sequence of events in menthol activation of TRPM8 channel. a** The free-energy landscape of a closed-to-open transition. Point mutations, depending on its location, have asymmetrical effects on the free energies of closed and open state, which can be reflected in the kinetics of single-channel events. **b** Representative single-channel recordings of WT and mutants. **c–f** Measured opening and closing rates (circles in blue and black, respectively) for point mutations at individual sites. For each mutation at the same site, 3-to-5 independent single-channel recordings were analyzed. **g–j** Brønsted plots to determine the Φ value as the slope of linear fitting (dashed line) for each site. All statistical data are given as mean ± s.e.m. **k** Φ values measured from single-channel recordings were mapped on the TRPM8 model. Residues moved early or late were colored in red or blue, respectively. Red and blue in color scale indicate relatively early and late movements, respectively. **l** $f_{progress}$ values calculated by iENM were mapped on the TRPM8 model. Residues moved early or late were colored in red or blue, respectively. **m** The Φ values (black) and $f_{progress}$ values (orange) matched well with each other at multiple residues. **n** A schematic diagram showing the temporal sequence of events upon menthol binding that lead to the full activation of TRPM8. All statistical data are given as mean ± s.e.m. Source data are provided as a Source Data file.

by the shifts in ANAP emission spectra. Conformational changes in the S1 to S4 domains open the S6 bundle crossing gate and induce conformational changes in near the selectivity filter to allow ion permeation (Fig. 6n).

Such a ligand-gating mechanism of TRPM8 channel shed new light on the investigation of voltage and temperature activation of this polymodal channel. Previous studies have shown that the positively charged R842 residue on S4 contributes to the total gating charge during voltage activation of the channel[11]. Our findings in this study and cryo-EM structures of TRPM8 show that menthol and its analog WS-12 interact with R842 to activate the channel[13], respectively. Since WS-12 is larger in its chemical structure than menthol (Supplementary Fig. 9a, b), the benzene ring moiety in WS-12 interacts with residues near the entrance of S1–S4 ligand-binding pocket so that WS-12 binds to a position lower than menthol (Supplementary Fig. 9d, e), while menthol lacking the benzene ring moiety could bind deeper inside the

pocket (Supplementary Fig. 9f). Therefore, these observations not only suggested that the ligand and voltage gating mechanisms are intrinsically coupled but also elucidated the R842 residue as the shared structural basis of such a coupling between two distinct stimuli. Moreover, though other TRPM8 channel agonist icilin (Supplementary Fig. 9c) and antagonists such as TC-I 2014 and AMTB, as well as the calcium ion, are different in chemical structures from menthol, cryo-EM studies have shown that they all bind to the ligand-binding pocket between S1 and S4 domain[13,14] (Supplementary Fig. 10), again illustrating that this domain serves as a hub for ligand binding.

Ligand, temperature, and voltage gating are also known to be allosterically coupled in TRPM8 channel[6,37]; however, how cold activates TRPM8 channel remains to be debated[38–40]. Our observation that menthol induces widely spread conformational changes within the transmembrane domains of the channel (Fig. 5f, g). Though we did not observe significant ANAP

emission shifts at the three sites on the TRP domain (Fig. 5f), it is possible that conformational changes could be reported at other sites on the TRP domain, but these mutants with ANPA incorporation were non-functional. Indeed, we tested nine ANAP-incorporation mutants on the TRP domain, but more than half of them (six out of nine) were non-functional (Supplementary Table 1). The importance of the TRP domain for ligand-gating in TRPM8 has been revealed by cryo-EM studies[12,13,19] and functional studies[10,41]. Such a pattern of conformational changes may also be relevant for cold activation of this channel, as theoretical analysis suggested that the structural elements involved in temperature gating of TRP channels may be scattered[42]. Understanding how each stimulus works will help unveil the complete gating mechanisms of TRPM8 channel.

In addition, we observed that menthol is a partial agonist for TRPM8 (Fig. 1c, Supplementary Fig. 1). This implies that because the channel protein still traverses between closed and open states even with saturating menthol, structural information of TRPM8 in menthol-activated state may be averaged out. For instance, the selectivity filter region remains missing in several cryo-EM structures (Supplementary Fig. 11a)[12–14]. This region has recently been determined in the calcium-bound desensitized state of parrot TRPM8 channel (PDB ID: 6O77)[14], with D907 and D908 pointing towards the pore (Supplementary Fig. 11b). Our computational model of mouse TRPM8 in the apo state[26] also suggested that corresponding D918 may face the central ion permeation pathway (Supplementary Fig. 11b, c). We reason that the selectivity filter conformation observed in cryo-EM structure[14] represents the structural state of the selectivity filter in a particular condition where the presence of calcium ions desensitized the channel. What we observed from the modeling represented another energetically stable state of the selectivity filter in the apo state without any ligand. Therefore, it is not surprising that selectivity filter showed distinct conformations in these two states. We acknowledge that the energetically stable conformation of the selectivity filter may not be the sole conformation this region may adopt, as the selectivity filter is structurally flexible.

Moreover, in our ANAP imaging experiments, shifts in ANAP emission spectrum cannot generate distance information between the fluorophore and residues of the channel protein to directly constrain structural modeling process. Though we have made our best efforts to control for factors such as temperature and pH that may affect ANAP fluorescence, there are other sources of uncertainties such as lipid composition of the membrane and the state of the channel-expressing HEK293 cells that further introduce ambiguity into the ANAP-guided modeling results. In addition, our thermodynamic mutant cycle analysis by itself also cannot derive the unambiguous atomic model of menthol-bound state. What the thermodynamic mutant cycle analysis provided us are three pairs of protein–menthol interactions (R842-hydroxyl group of menthol, I846-isopropyl group of menthol, and L843-isopropyl group of menthol). Based on these three pairs of interactions, we know how menthol molecule locally binds within its binding pocket. This is still far away from any unambiguous atomic model of menthol-bound state of the intact TRPM8 channel, and we are not claiming that we have got such a model from thermodynamic mutant cycle analysis and ANAP imaging experiments. Therefore, a model of TRPM8 channel in the menthol-bound open state based on cryo-EM or crystallography is still needed in future.

Revealing the ligand-gating mechanism of TRPM8 channel is of clear translational merits. In tobacco industry, menthol has been widely used as an additive in cigarettes to soothe the irritation of smoke[43]. Understanding how menthol activates TRPM8 will lead to better regulation and manufacturing of menthol-added tobacco products to minimize health hazards. Moreover, TRPM8 is critically involved in prostate cancer not only because

it is highly expressed in prostate epithelium and prostate cancer cells, but also the hormone testosterone works as an endogenous ligand to bind and activate this channel[44–46]. In addition, chemotherapy against cancer often induces cold allodynia mediated by TRPM8 channel[47], so its antagonists would be beneficial in cancer research and treatment. As current TRPM8 inhibitor like PF-05105679 exhibit side effects of altering temperature sensation in clinical trials[8,47], understanding the ligand-gating mechanism of this channel in detail will help develop novel therapeutics targeting TRPM8 with less adverse effects in future.

## Methods

**Molecular biology and cell transfection.** Murine TRPM8 was used in this study. Point mutations were made by Fast Mutagenesis Kit V2 (SBS Genetech). Primers used to generate point mutations are summarized in Supplementary Table 2. All mutations were confirmed by sequencing.

HEK293T cells purchased from Kunming Cell Bank, Kunming Institute of Zoology, Chinese Academy of Sciences (ATCC, CRL-3216) were cultured in Dulbecco's modified Eagle's medium supplemented with 20 mM L-glutamine and 10% fetal bovine serum. Cells were transiently transfected with cDNA constructs by Lipofectamine 3000 (Life Technologies) following the manufacturer's protocol. Patch-clamp recordings were performed 1–2 days after transfection.

**Chemicals.** (−)-Menthol was purchased from BBI Life Sciences (CAS: 2216-51-5); p-menthane was purchased from Wako Pure Chemical Industries (CAS: 99-82-1); (−)-menthone and 3-methylcyclohexanol were purchased from Tokyo Chemical Industry (CAS: 14073-97-3 and 591-23-1, respectively); cyclohexanol was purchased from Sangon Biotech (CAS: 108-93-0).

**Fluorescence unnatural amino acid.** L-ANAP methyl ester was purchased from AsisChem. pANAP vector was purchased from Addgene. ANAP was incorporated into TRPM8 by introducing a TAG amber stop codon mutation as previously reported[30]. Briefly, after co-transfection of both TRPM8 and pANAP vectors, ANAP was directly added to the culture medium to the final concentration of 20 μM. After 1–2 days, ANAP-containing culture medium was completely changed. Cells were further cultured in ANAP-free medium overnight before experiments.

ANAP fluorescence was excited by the Ar laser with a 375/28 excitation filter, a T400lp dichroic mirror, and a 435LP emission filter on an inverted fluorescence microscope (Nikon TE2000-U) using a ×40 oil-immersion objective (NA 1.3). Emission spectrum of ANAP was imaged with an Acton SpectraPro 2150i spectrograph in conjunction with an optiMOS CCD camera (Qimaging, USA). We determined the ANAP emission peak value by fitting the emission spectrum with a skewed Gauss distribution in Igor Pro version 6.1 (WaveMatrix), and then the difference in emission peak values measured in the absence and presence of menthol was reported as the shift of ANAP emission.

**Electrophysiology.** Patch-clamp recordings were performed with a HEKA EPC10 amplifier controlled by PatchMaster software (HEKA). Whole-cell recordings were performed at ±80 mV. Patch pipettes were prepared from borosilicate glass and fire-polished to resistance of ~4 MΩ. For whole-cell recording, serial resistance was compensated by 60%. For single-channel recordings, patch pipettes were fire-polished to a higher resistance of 6-to-10 MΩ and then recorded in the inside-out configuration. To maximize the chance of obtaining a patch with only one channel, single-channel recordings were performed about 8 h after transfection. A solution with 130 mM NaCl, 10 mM glucose, 0.2 mM EDTA, and 3 mM HEPES (pH 7.2) was used in both bath and pipette for either whole-cell or single-channel recordings. Membrane potential was clamped at +80 mV for single-channel recordings. Current was sampled at 10 kHz and filtered at 2.9 kHz. All recordings were performed at room temperature (22 °C) with the maximum variation of 1 °C.

Ligands such as menthol and its analogs were perfused to membrane patch by a gravity-driven system (RSC-200, Bio-Logic). Bath and ligand solution were delivered through separate tubes to minimize the mixing of solutions. Patch pipette was placed in front of the perfusion tube outlet.

**Φ-analysis.** We followed the principle of Φ-analysis described in published literatures in detail[17,36,48]. To ensure saturation of binding in our experiments, 5 mM menthol was used. At such a high concentration, only the patches showing the activities of one channel was used for analysis. Single-channel data were processed by the QuB software[49] version 1.4 (ref. [50]). Opening and closing events were detected during idealization with the half amplitude method. Using the simple close ↔ open model (Fig. 6a), the forward and backward rates were determined with a deadtime of 0.3-to-0.4 ms. The equilibrium constant was calculated as the ratio of forward and backward rates. Opening rates and equilibrium constants for each mutant at the same residue site were plotted in log scale (Brønsted plot) before fitting with a linear function to determine the Φ value.

**Data analysis**. Data from whole-cell recordings were analyzed in Igor Pro version 6.1 (WaveMatrix). $EC_{50}$ values were derived from fitting a Hill equation to the concentration–response relationship. Changes in $EC_{50}$ by point mutation may be caused by either perturbation of ligand bind or channel gating or both. To distinguish these possibilities, dissociation constant ($K_d$) for ligand binding was estimated assuming the following gating scheme:

$$C_0 \overset{K_d}{\leftrightarrow} C_1 \overset{L}{\leftrightarrow} O, \tag{1}$$

where $L$ is the equilibrium constant for the final closed-to-open transition.

To perform thermodynamic cycle analysis, $K_d$ values of four channel-ligand combinations (WT channel, menthol: $K_d\_1$; Mutant channel, menthol: $K_d\_2$; WT channel, menthol analog: $K_d\_3$; Mutant channel, menthol analog: $K_d\_4$) were determined separately. The strength of coupling was determined by the coupling energy (kT multiplied by $Ln\Omega$, where $k$ is the Boltzmann constant and $T$ is temperature in Kelvin). $Ln\Omega$ was calculated by Eq. (2) as[51]

$$Ln\Omega = Ln\left(\frac{K_d\_1 \cdot K_d\_4}{K_d\_2 \cdot K_d\_3}\right). \tag{2}$$

From single-channel recordings, the open probability ($P_o$) was calculated after events detection with a hidden Markov approach in the software QuB[49] (http://www.qub.buffalo.edu/). A dead time of 0.32 ms was imposed. As a complementary approach, $P_o$ was also estimated from noise analysis[52] of whole-cell recordings[53]. Briefly, the mean current amplitude ($I$), the squared deviations in current amplitude from the mean value ($\sigma^2$), and the single-channel current ($i$) were measured experimentally from a membrane patch of ion channels. Then the number of ion channels clamped in that patch ($N$) was determined as

$$N = \frac{I^2}{i \cdot I - \sigma^2}. \tag{3}$$

The maximum current when each of ion channel is at the open state with a $P_o$ of 1 is equal to $i \times N$. Then the open probability was calculated as the ratio between the measured macroscopic current and the maximum current calculated by noise analysis. To estimate channel conductance, current amplitude was estimated from all-point histogram of single-channel recordings.

**Molecular modeling**. To model the missing selectivity filter of TRPM8 in the apo state, membrane-symmetry-loop modeling was performed using the Rosetta molecular modeling suite[54] version 2016.20. Starting with the cryo-EM structure of the apo state (PDB ID: 6BPQ), the selectivity filter, the pre-S6 linker, and the S1–S2 linker were modeled de novo with the kinematic (KIC) loop modeling protocol[55,56]. In total, 10,000–20,000 models were generated each round. After nine rounds of KIC loop modeling, the top 10 models converged well. The model with the lowest energy was finally selected as the open state model. This model was further refined by the relax application[57] within the Rosetta suite.

To model the potential menthol-induced open state of TRPM8, a similar modeling procedure was employed. The S1–S4 domain (residues 734–861) and the TRP domain (residues 992–1013) were modeled using the corresponding domains in the cryo-EM structure of WS-12 bound state (PDB ID: 6NR2) as templates for homology modeling in Rosetta. For the channel pore, S5 (residues 862–889), pore helix (residues 898–909) and S6 (residues 951–979) were modeled using the corresponding domains in the cryo-EM structure of WS-12 bound state (PDB ID: 6NR2) as templates for homology modeling in Rosetta. For loop regions between the helical domains above (S5-pore helix loop: residues 890–897; pore helix-S6 loop: residues 910–950; S6-TRP domain loop: residues 980–991), they were modeled with KIC loop modeling in Rosetta. After each round of KIC modeling, the models were first filtered by changes in SASA values measured from ANAP imaging experiments. An increase in ANAP emission peak indicates that the amino acid sidechain at the incorporated site transits from hydrophobic to hydrophilic environment[30], which is accompanied by an increase in solvent accessible surface area (SASA) of the sidechain[17]. SASA can be directly measured from a protein structure, so we can impose changes in SASA during computational modeling to filter out the models that are inconsistent with ANAP imaging results. Previously we have used such methods to model the capsaicin-induced open state of TRPV1 channel[17]. However, shifts in ANAP emission spectrum cannot generate distance information between the fluorophore and residues of the channel protein to directly constrain structural modeling process. As we did not observe ANAP emission shifts in the TRP domain and relatively small changes in ANAP in the S1–S4 domain, we constructed model of these two domains of mouse TRPM8 based on the cryo-EM structure of TRPM8 in the menthol analog WS-12 bound state (PDB ID: 6NR2) using homology modeling. With increase in SASA at A875, D920, L939, P958, and S966 and decrease in SASA at T922 as filtering constraints, only the models with an increase in SASA larger than 10 Å2 compared to the closed state were allowed to pass. Among the filtered models, the top 20 models by energy was selected as the inputs for next round of loop modeling. After six rounds of KIC loop modeling, the top ten models converged well (Supplementary Fig. 5). The model with the lowest energy was finally selected as the open state model. This model of the pore was combined with the separately built model of S1–S4 domain and the TRP domain, and then further refined by the relax application[57] within the Rosetta suite.

By comparing our potential models of TRPM8 channel in the closed and open states (Supplementary Fig. 6a, b, respectively), we clearly observed that the residues employed as constraints exhibited SASA changes consistent with the ANAP emission shifts, as residues showing positive ANAP shifts also exhibit increase in SASA and vice versa (Supplementary Fig. 6c). The right shifts in ANAP emission measured at sites 774 and 791 served as supporting data for our computational modeling of menthol-activated open state. We observed that the SASA changes calculated from our models at sites 774 and 791, which were not included in the model-building process, also agree with the ANAP shifts measured experimentally (Supplementary Fig. 6d, data points in purple).

When we compared the TRPM8 channel pore, we observed that in the open model both the selectivity filter and the S6 bundle crossing were wide enough to allow ions and water molecules to pass (Supplementary Fig. 7a, b, the model in cyan). We further calculated the conductance of our TRPM8 models using the HOLE program[58]. Both the predicted conductance of this model and our measured conductance from single-channel recordings were much larger than the value predicted for our TRPM8 model in the closed state (Supplementary Fig. 7c).

Command lines used in Rosetta to perform the modeling processes were attached in Supplementary Methods. SASA of each residue in TRPM8 structures models was measured by RosettaScripts[21] within the Rosetta suite. The scripts to perform SASA measurements and filtering were also attached in Supplementary Methods.

Pore radius of a TRPM8 model was calculated by the HOLE program[59] version 2.0 (ref. [58]).

All the molecular graphics of menthol and TRPM8 were rendered by UCSF Chimera[60] software version 1.12.

**Molecular docking**. RosettaLigand application[22,61,62] from Rosetta program suite version 2016.20 was used[54] to dock ligand to TRPM8. We docked menthol into the binding pocket formed by S1–S4 domain (residues 734–861) and TRP domain (residues 992–1013) in the potential open state model of TRPM8 built as described in "Molecular modeling" above. TRP domain is important for ligand gating in TRPM8 as revealed by cryo-EM studies[12,13,19] and functional studies[10,41], so this domain is included in our docking experiments. The model was then relaxed in membrane environment using the RosettaMembrane application[20,63,64] and the model with lowest energy scores were chosen for docking of menthol. For docking of WS-12, icilin, AMTB, and TC-l 2014, similar to docking of menthol, the transmembrane domains from beginning of S1 to the end of TRP domain in cryo-EM structure of TRPM8 in the WS-12 bound state (PDB ID: 6NR2), icilin bound state (PDB ID: 6NR3), AMTB bound state (PDB ID: 6O6R), and TC-l 2014 bound state (PDB ID: 6O72) was first relaxed in membrane environment using the RosettaMembrane application[20,63,64], respectively. The models with lowest energy scores were chosen for docking of each ligand.

Docking comprised three stages, which progressed from low-resolution conformational sampling and scoring to full atom optimization using all-atom energy function. In the first, low-resolution stage, menthol (or WS-12, icilin, AMTB and TC-l 2014) molecule was initially placed roughly in the center of the cavity defined by S1 to S4 facing the cytosol. Its "center of mass" was constrained to move within a 10 Å diameter sphere, where it could move freely during the docking process. Menthol (or WS-12, icilin, AMTB, and TC-l 2014) conformers were generated using the FROG2 server[65] (http://mobyle.rpbs.univ-paris-diderot.fr/cgi-bin/portal.py#forms::Frog2). The second, high-resolution stage employed the Monte Carlo minimization protocol in which the ligand position and orientation were randomly perturbed by a small deviation (0.1 Å and 3°); channel residue side chains were repacked using a rotamer library; the ligand position, orientation, and torsions and protein sidechain torsions were simultaneously optimized using quasi-Newton minimization and the end result was accepted or rejected based on the Metropolis criterion. The sidechain rotamers were searched simultaneously during "full repack" cycles and one at a time in the "rotamer trials" cycles. The full repack made ~$10^6$ random rotamer substitutions at random positions and accepted or rejected each based on the Metropolis criterion. Rotamer trials chose the single best rotamer at a random position in the context of the current state of the rest of the system, with the positions visited once each in random order. The ligand was treated as a single residue and its input conformers served as rotamers during this stage. The third and final stage was a more stringent gradient-based minimization of the ligand position, orientation, and torsions and the channel torsion angles for both side chains and backbone.

As menthol (or WS-12, icilin, AMTB, and TC-l 2014) binds to the transmembrane region of TRPM8, the molecular docking approach must consider the energetic effects of the lipid membrane. The membrane environment was setup using RosettaMembrane energy function[20,63,64] in a XML style script in RosettaScripts[21] (Supplementary Methods). The script also allowed us to control the details of docking. A total of 10,000 models were generated for a docking trial. To determine the best docking model, these models were first screened with total energy score (Rosetta energy term name: *score*). Top 1000 models with lowest total energy score were selected. They were further scored with the binding energy between menthol and the channel. Binding energy was calculated as the difference in total energy between the menthol-bounded state and the corresponding apo state models. Top 10 models with lowest binding energy (*interface_delta_X*) were identified as the candidates.

To reveal the spatial distribution of binding interaction between menthol and the channel, VDW and hydrogen bond energies were further mapped on a per-residue basis to the channel by Rosetta's residue_energy_breakdown utility. Average values of VDW energy and hydrogen bond energy were calculated based the top 10 models with the lowest binding energy.

**Elastic network modeling**. iENM was performed using iENM web server[34] (http://enm.lobos.nih.gov). Transmembrane domains of TRPM8 channel in the closed state structure and potential menthol-induced open state were submitted as the starting and ending conformation, respectively. The distance cutoff for elastic interaction between alpha carbon atoms was set as 15 Å. Based on this cutoff, two harmonic potentials were constructed for the starting and ending conformations, respectively. The server solved the saddle points of a general potential functions composed of these two harmonic potentials. The calculated $f_{progress}$ values reflected the temporal sequence of movements.

**ConSurf analysis**. TRPM8 protein sequence conservation analysis was performed using ConSurf web server[18] (http://consurf.tau.ac.il/). Our models of TRPM8 transmembrane domains in the closed and open states were submitted to the server with default parameters. The results of ConSurf analysis was visualized in the UCSF Chimera.

**Statistics**. All experiments have been independently repeated for at least three times. All statistical data are given as mean ± s.e.m. Two-sided Student's $t$-test was applied to examine the statistical significance. NS indicates no significance. *, **, and *** indicate $p < 0.05$, $p < 0.01$, and $p < 0.001$, respectively.

**Reporting summary**. Further information on research design is available in the Nature Research Reporting Summary linked to this article.

## Data availability

Data supporting the findings of this manuscript are available from the corresponding authors upon reasonable request. A reporting summary for this Article is available as a Supplementary Information file. The source data underlying Figs. 1b, h, 2d–g, 3c, 3e–g, 4b–e, 5f, 6k, m and Supplementary Fig. 6c, Supplementary Fig. 6d and Supplementary Fig. 7c, as well as our model of menthol docking, the potential model of TRPM8 in the open states (Supplementary Data 1), are provided as the Source Data Files.

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

## Acknowledgements

We are grateful to our lab members for assistance. We thank Dr. Jie Zheng for insightful discussions. This work was supported by funding from National Key R&D Program of China (2017YFA0103700), the National Natural Science Foundation of China (31571528), the National Natural Science Foundation of Zhejiang Province (LR15H020001) to P.L.; from the National Major New Drugs Innovation and Development (No. 2018ZX09711001-004-005), the Zhejiang Provincial Natural Science Foundation of China (Nos. LR16H090001), the Natural Science Foundation of China (81571127) to W.Y.; from the American Heart Association (14GRNT18980033 and 17GRNT33690009) to W.Z.; from the National Science Foundation of China (31640071 and 31770835), Chinese Academy of Sciences (Youth Innovation Promotion Association and "Light of West China" Program), and Yunnan Province (2017FB037 and 2018FA003) to S.Y., and from National Science Foundation of China (31741067 and 31800990) and Zhejiang Provincial Natural Science Foundation of China (LR20C050002) to F.Y. This work was also supported by the Core Facilities in Zhejiang University School of Medicine, including the Olympus FV3000 fluorescence imaging microscope and the bioinformatics computation platform. Molecular dynamic simulation support was provided by the Center for Computational Research at the University at Buffalo and the Biowulf system at the National Institutes of Health.

## Author contributions

L.X., Y.H., X.C., A.A., H.W., X.L., and Z.Z. conducted the experiments including patch-clamp recording, mutagenesis, imaging, molecular modeling, and data analysis; H.W. and W.Z. performed iENM simulations; S.Y. and F.Y. prepared the manuscript; C.M., P.L., W.Y., S.Y., and F.Y. conceived and supervised the project, participated in data analysis and manuscript writing.

## Competing interests

The authors declare no competing interests.
