## [Peer Review File · Nature Communications]

Reviewers' comments first round:

Reviewer #1 (Remarks to the Author):

Fan Yang's group examined structural changes of menthol-dependent gating of TRPM8 channel. Computational docking of the menthol to the channel was performed, followed by thermodynamic mutant cycle analysis with mutated channels and derivatives of menthol (p-menthane and menthone). This shows grab and stand profile dependent on hydrogen bond and VDW interaction. Then, conformational change of individual amino acids of the channel was surveyed by fluorometry using environment-sensitive fluorescent amino acid, ANAP. By taking these results of ANAP measurements into account as structural restrictions, structural model of menthol-activated channel was constructed. They also performed elastic network modeling using the web server and obtained results of temporal sequence of movements, consistent with the idea of motions of two gates (earlier motion at the S6 bundle crossing and later motion at the selectivity filter) which follow the structural changes of the menthol binding pocket and the S4-S5 linker. This study is elegantly designed and distinct disciplines were impressively well integrated toward understanding gating of this interesting channel, including electrophysiology, fluorometry, and mathematical calculation. This is high quality, comprehensive study on menthol docking structure and structural change underlying channel gating of TRPM8.

I have the following concerns.

Major points:

1. Nothing is said for ANAP fluorometry results of S1-S3. I wonder if ANAP signal also changes similarly in other transmembrane segments, such as S1, S2 and S3.

2. ANAP results were elegantly incorporated into computational modeling of the menthol-gated channel. This is based on an idea that a shift of ANAP spectrum exclusively depends on hydrophobicity-hydrophilicity change of environment. I agree that ANAP fluorescence spectrum is shifted dependent on environmental shift of hydrophobicity/hydrophilicity. However, I guess, whether shift of ANAP fluorescence spectrum in proteins occurs only upon environmental change of hydrophobicity/hydrophilicity is still not established. I recommend that authors will argue about a possibility that other physicochemical basis than hydrophobicity-hydrophilicity transition might cause shift of fluorescence spectrum.

3. Evidence for authors' conclusion of later motion of selectivity filter than the motion of S6 bundle crossing relies only on results of iENM (color drawing in Fig5e). Readers who are not familiar with iENM analysis, like me, will not be convinced just by seeing color pattern like Fig5e (even though the conclusion is predictable from gating mechanisms of other 6TM channels). Any quantitative data which support the result of Fig5e would be demonstrated, if possible. Related to this concern, have authors performed rate-equilibrium linear free-energy relationships, as author's group has previously done Yang F et al. Nature Communi.(2018) 9:2879 in their analysis of TRPV1 channel in parallel with iENM, although this analysis may require robust and rigorous analysis of single channel currents?

Minor points:

1. It would be more clearly stated how shift of ANAP spectrum was measured. Was simply the peak difference measured, or was it measured in a different way?

2. In Fig5e (iENM data), some part of the voltage sensor domain like region (S1-S4) has red color, suggesting that VSD-like region continues to rearrange structure (perhaps even after gating occurs?). As far as I see the paper, I could not find any statement on "late motion" of the VSD-like region.

Reviewer #2 (Remarks to the Author):

Molecular mechanisms underlying menthol binding and activation of TRPM8 ion channel

Xu et al.

A molecular structure of TRPM8 channel in complex with the naturally occurring agonist menthol is currently unavailable and the mechanism by which menthol activates TRPM8 has been elusive to date. To study the mechanisms of menthol binding and activation of TRPM8 channel, Xu et al. performed computational modeling combined with mutagenesis and functional characterization and proposed a menthol bound structural model as well as the menthol-dependent conformational changes in TRPM8 that may lead to the channel opening. The structural and mechanistic models proposed by the authors lack novel insights and cannot be substantiated by their experimental data. More critically, the results contradict those from the previous functional and structural studies of TRPM8, which authors have failed to address in their manuscript. Overall, the results from computational and functional studies are somewhat forced and do not offer significant insights into TRPM8 function.

Major comments

1) High-throughput screening and mutagenesis study (Bandell et al., 2006) revealed residues Y745 and Y1005 are required for TRPM8 activation by menthol. Additionally, a cryo-EM structure of TRPM8 in complex with the menthol analog WS-12 (Yin et al., 2019) shows that the menthol-like moiety in WS-12 is sandwiched between Y745 and Y1005 and interacts with these two residues, supporting the predictions from the high-throughput screening study. This structure suggested that menthol binds to TRPM8 in a way similar to the menthol-like moiety in WS-12. However, the menthol binding site identified by the authors from their docking model is distinct from where the menthol-like moiety in WS-12 is positioned in the cryo-EM structure; the menthol molecule is not capped by Y745 but is located above the WS-12 binding site and more importantly, Y745 and Y1005 do not form prominent interactions with the menthol molecule, which is inconsistent with the previously reported functional studies by Bandell et al. Therefore, the binding configuration of menthol depicted in this study is questionable.

2) The menthol-induced conformational changes described in this study are contradictory to the agonist-induced structural rearrangements shown in the recently published TRPM8 complex structures. By comparing cryo-EM structures of TRPM8 in the apo and in the agonist-bound states, Yin et al. showed that (1) the binding of the menthol analog WS-12 or the synthetic super-cooling agent icilin triggers an outward rotation of the VSLD away from the pore domain; (2) the binding of the allosterically coupled PIP2 and agonist would induce a bending in the S5 helix. However, in this study, the authors identified that binding of menthol at R842 in S4 induced inward movement of S2 and S3 towards S1 and S4, followed by an outward movement of the S4-S5 linker and opening of the S6 gate. The menthol-induced conformational changes based on the computational modeling are questionable, so the authors cannot suggest coupling between menthol and voltage in TRPM8 gating using these results.

3) I realized that instead of using the WS-12 and PIP2-bound TRPM8 structure as a template, Xu et al. chose to use the apo TRPM8 structure. It is a well-established fact that TRPM8 requires the membrane lipid PI(4,5)P2 for the channel activation and depletion of PI(4,5)P2 leads to channel desensitization (or rundown). Furthermore, menthol-mediated TRPM8 activation is allosterically modulated by PI(4,5)P2, and the binding of menthol and PI(4,5)P2 is allosterically coupled. The computational studies for menthol docking as well as their open state model did not include PI(4,5)P2 but the computational studies led to an open state, further calling into a question the results from modelling studies.

4) I realized that the molecular docking approach is biased, because the authors initially placed the menthol molecule roughly in the center of the cavity defined by S1 to S4, rather than using the full channel and starting from a random site.

5) I realized that authors did omit the comparison of their structural and mechanistic models with recent complex structures of TRPM8 by the Lee group as well as previous functional data by the Patapoutian group. The authors should make comparisons and discuss the differences.

6) Based on the docking model, to validate specific interactions between residues in the TRPM8 VSLD cavity and menthol functional groups, the authors performed thermodynamic mutant cycle analysis between VSLD mutants and menthol analogs. The caveat for this approach is the use of double-mutant cycle analysis between protein and (a very) small molecule. Double mutant cycle analysis is a useful tool when applied to protein-protein interactions, but not so much for the small molecule-protein interaction because removing or mutating a functional moiety in the small molecule often leads to a completely different interaction profile with protein. In particular, menthol exhibits a relatively small chemical structure. The authors removed its hydroxyl or

isopropyl group and measured the apparent binding affinity to the wildtype or mutant channel. Binding of these modified menthol analogs might bind nonspecifically to a location elsewhere in TRPM8. In this case using double-mutant cycle analysis to probe interactions between menthol and residues in VSLD lacks legitimacy. Although the Zheng group used the same approach to study the mechanisms of capsaicin binding and TRPV1 channel activation, the double-mutant cycle analyses between capsaicin and TRPV1 channel were systematically designed. Capsaicin is larger than menthol and contains amphipathic nature. Furthermore, key functional groups other than aliphatic tail in capsaicin were systematically replaced but not removed, which preserve its interaction profile with TRPV1.

7) Authors computationally determined TRPM8 structure in the closed state and employed ANAP imaging to reveal residues that showed conformational changes induced by menthol binding. The ANAP fluorescence data were incorporated for computational modeling of the menthol-induced open state, where the changes in the solvent accessible surface area (SASA) of the side chain were imposed as constraints. However, ANAP only provides information on accessibility change in a qualitative manner, but it does not provide information (e.g. distance) that can be used as constraints for modeling. It is unclear how ANAP data can be incorporated into modeling an open state. For example, the large ANAP signal cannot be interpreted as large conformational changes and can be interpreted in multiple ways. As far as I know there are no systematic studies to correlate ANAP signals for structural changes. Residues showing ANAP signal shifts are located at the ion permeation pathway-lining region and thus it is difficult to draw any meaningful mechanistic insights.

Minor comments

- 1) In methods, line 420 and 423, the authors wrote "dock capsaicin to TRPM8". It should be "menthol".
- 2) Labels for residues and helices are missing in Figure 1f-g.
- 3) In Fig. S1, the single-channel current traces are too noisy and appear to have multiple subconductance.

Reviewer #3 (Remarks to the Author):

Summary of paper:

Experiments and computations are used to suggest a mechanism of how menthol binds and activates its receptor: the TRPM8 ion channel. The computational methodology includes structural modeling and docking, as well as elastic network model analysis of the equilibrium dynamics, and the experimental techniques include mostly electro physiology, double-mutant cycles, and using unnatural fluorescence amino acid.

Opinion:

The problem is interesting and important, and the approach suitable. The interplay between computation and experiment helps to decipher the activation mechanism in great details. The first part, with menthol docking is very convincing; the stereochemistry of the binding pose makes sense. The second part, where elastic network analysis is used to suggest the consequent conformational changes in the channel is a bit less convincing. Overall, it is a very good manuscript that is suitable for publication after relatively minor changes.

Major issues:

1. A bit more work is needed to further support the suggested conformational changes in TRPM8 upon menthol binding. Here are two suggestions in this direction (but there could be better ideas): (a) the elastic network model could reveal hinge residues that are important for the conformational changes. These could be examined in mutagenesis. (b) Evolutionary conservation analysis, e.g., using ConSurf (<http://consurf.tau.ac.il>) could be used to highlight key amino acids that are important for the mechanism. It would be helpful to correlate these with the observations from the dynamics.

2. Modelling the selectivity filter in a closed and open states: Apparently, the authors have an entire paper about this modelling project but it is still in press. From what they describe here, they used Rosetta de novo modelling. Since loop modelling is tricky, it is worthwhile to try some other methods and compare the results. For example, the TRPM2 channel has a 45% sequence identity to the TRPM8 channel. TRPM2 structure was solved in both the apo-closed and Ca²⁺-bound open states (6MIX and 6MJ2 respectively). It seems that TRPM2 structure includes the loop regions that are missing here. This might be a better starting point for modelling both the closed and open states compared to de novo modelling. In addition, it would be helpful to run ConSurf and check that the amino acids facing the filter are the most highly conserved, as they should. The ConSurf analysis could also help select the most suitable model of the loops among various decoys.

Minor issues:

1. Paragraph starting in line 57: A figure/supplementary figure showing the different domains and the nomenclature of the helices could help readers that are not familiar with this channel.
2. Line 144: what is Fig. 1s?
3. Line 420 and line 423: Presumably menthol rather than capsaicin.
4. Figure 1 f-g, amino acids numbering is missing.
5. General comments concerning the figures: Jamming 8 different panels into one figure can be confusing.
6. The manuscript reads very well overall, but a few sentences could be further edited.

Nir Ben-Tal

Reviewers' comments:

Reviewer #1 (Remarks to the Author):

Fan Yang's group examined structural changes of menthol-dependent gating of TRPM8 channel. Computational docking of the menthol to the channel was performed, followed by thermodynamic mutant cycle analysis with mutated channels and derivatives of menthol (p-menthane and menthone). This shows grab and stand profile dependent on hydrogen bond and VDW interaction. Then, conformational change of individual amino acids of the channel was surveyed by fluorometry using environment-sensitive fluorescent amino acid, ANAP. By taking these results of ANAP measurements into account as structural restrictions, structural model of menthol-activated channel was constructed. They also performed elastic network modeling using the web server and obtained results of temporal sequence of movements, consistent with the idea of motions of two gates (earlier motion at the S6 bundle crossing and later motion at the selectivity filter) which follow the structural changes of the menthol binding pocket and the S4-S5 linker. This study is elegantly designed and distinct disciplines were impressively well integrated toward understanding gating of this interesting channel, including electrophysiology, fluorometry, and mathematical calculation. This is high quality, comprehensive study on menthol docking structure and structural change underlying channel gating of TRPM8.

I have the following concerns.

Major points:

1. Nothing is said for ANAP fluorometry results of S1-S3. I wonder if ANAP signal also changes similarly in other transmembrane segments, such as S1, S2 and S3.

Response:

We have performed additional ANAP imaging experiments for residues on the S1-S3 domains as suggested by the reviewer. There we found eight sites (734, 774, 787, 789, 791, 795, 808 and 817) that were tolerant to ANAP incorporation (Fig. 5f). Among these eight sites, ANAP incorporated at site 774 of the S2 and 791 of the S3 exhibited significant right shifts larger than 2 nm in emission spectra during menthol activation of the channel. Such shifts in ANAP emission indicate that the S2 and S2-S3 linker undergo conformational changes during menthol activation.

More importantly, the right shifts in ANAP emission measured at sites 774 and 791 serve as validating data for our computational modeling of menthol-activated open state. Our current model was built with ANAP constraints at sites 824, 875, 920, 922, 939, 958 and 966 (Fig. 5 and Fig. 6), so it is expected that the SASA changes calculated at these model-building constraining sites are fully consistent with the shifts in ANAP measured (Fig. S5c). We observed that the SASA changes calculated from our models at sites 774 and 791, which were not included as constraints in the model-building process, also agree with the ANAP shifts measured experimentally

(Fig. S5d, data points in purple). Therefore, we believe that our additional ANAP measurements have not only validated our computational models, but also revealed conformational changes induced by menthol binding (Fig. 6). We have modified the manuscript accordingly.

2. ANAP results were elegantly incorporated into computational modeling of the menthol-gated channel. This is based on an idea that a shift of ANAP spectrum exclusively depends on hydrophobicity-hydrophilicity change of environment. I agree that ANAP fluorescence spectrum is shifted dependent on environmental shift of hydrophobicity/hydrophilicity. However, I guess, whether shift of ANAP fluorescence spectrum in proteins occurs only upon environmental change of hydrophobicity/hydrophilicity is still not established. I recommend that authors will argue about a possibility that other physicochemical basis than hydrophobicity-hydrophilicity transition might cause shift of fluorescence spectrum.

We agree with the reviewer that besides the hydrophobicity in the local chemical environment, other physio-chemical factors such as temperature and pH are known to affect excitation/emission properties of a fluorophore. Specifically for ANAP, which is an amino acid derivative of PRODAN, its absorption spectra can be shifted from 362nm at 24°C to 366nm at 4°C (Macgregor R.B. and Weber G. Nature 1986). Therefore, we performed our ANAP experiments always at room temperature ($22 \pm 1^\circ\text{C}$). To control for changes in pH, we used 3 mM Hepes in ANAP imaging solution to buffer the pH at 7.2. Therefore, we believe that the shifts in ANAP emission during menthol activation of TRPM8 channels are mostly caused by changes in hydrophobicity induced by protein conformational rearrangements. We have added more discussion on the causes of ANAP emission shifts in the manuscript at line 259-262.

3. Evidence for authors' conclusion of later motion of selectivity filter than the motion of S6 bundle crossing relies only on results of iENM (color drawing in Fig5e). Readers who are not familiar with iENM analysis, like me, will not be convinced just by seeing color pattern like Fig5e (even though the conclusion is predictable from gating mechanisms of other 6TM channels). Any quantitative data which support the result of Fig5e would be demonstrated, if possible. Related to this concern, have authors performed rate-equilibrium linear free-energy relationships, as author's group has previously done Yang F et al. Nature Communi.(2018) 9:2879 in their analysis of TRPV1 channel in parallel with iENM, although this analysis may require robust and rigorous analysis of single channel currents?

We agree with the reviewer that to reveal the temporal sequence of events induced by menthol to open TRPM8 channel, both computational and functional experiments are needed, so we have performed the rate-equilibrium linear free-energy relationship analysis (Φ analysis) with single-channel recordings. Due to the limitation of time and resources, we generated 47 point mutations on the TRPM8 channel, which enabled us

to perform Φ analysis on four sites (840 in the ligand binding pocket, 910 and 916 near the upper gate, 972 near the lower gate) (Fig. 7). We observed the largest Φ value at site 840, and sites near the upper gate shows smaller Φ values than that near the lower gate. Such a distribution of Φ values indicate the ligand binding pocket and the lower gate move first upon menthol application, while the upper gate near the selectivity filter opens later. Our experimentally measured Φ values correlate well with the f_{progress} values calculated from our structural models by the iENM method (Fig. 8), consolidating our proposed temporal sequence of channel-activation events upon menthol binding.

Minor points:

1. It would be more clearly stated how shift of ANAP spectrum was measured. Was simply the peak difference measured, or was it measured in a different way?

We determined the ANAP emission peak value by fitting the spectrum with a skewed Gauss distribution, and then the difference in emission peak values measured in the absence and presence of menthol was reported as the shift of ANAP spectrum. We have modified the method session to clarify this point.

2. In Fig5e (iENM data), some part of the voltage sensor domain like region (S1-S4) has red color, suggesting that VSD-like region continues to rearrange structure (perhaps even after gating occurs?). As far as I see the paper, I could not find any statement on “late motion” of the VSD-like region.

We agree with the reviewer that iENM analysis indicated late motion in the VSD-like domain. As pointed out by the reviewer, such late motion may reflect the continuing movements of channel protein to accommodate conformational rearrangements in the upper and lower gates opened by menthol binding. We discuss on this point in the manuscript around line 375-379.

Reviewer #2 (Remarks to the Author):

Molecular mechanisms underlying menthol binding and activation of TRPM8 ion channel

Xu et al.

A molecular structure of TRPM8 channel in complex with the naturally occurring agonist menthol is currently unavailable and the mechanism by which menthol activates TRPM8 has been elusive to date. To study the mechanisms of menthol binding and activation of TRPM8 channel, Xu et al. performed computational modeling combined with mutagenesis and functional characterization and proposed a menthol bound structural model as well as the menthol-dependent conformational changes in TRPM8 that may lead to the channel opening. The structural and mechanistic models proposed by the authors lack novel insights and cannot be substantiated by their experimental data. More critically, the results contradict those from the previous functional and structural studies of TRPM8, which authors have failed to address in their manuscript. Overall, the results from computational and functional studies are somewhat forced and do not offer significant insights into TRPM8 function.

Major comments

1) High-throughput screening and mutagenesis study (Bandell et al., 2006) revealed residues Y745 and Y1005 are required for TRPM8 activation by menthol. Additionally, a cryo-EM structure of TRPM8 in complex with the menthol analog WS-12 (Yin et al., 2019) shows that the menthol-like moiety in WS-12 is sandwiched between Y745 and Y1005 and interacts with these two residues, supporting the predictions from the high-throughput screening study. This structure suggested that menthol binds to TRPM8 in a way similar to the menthol-like moiety in WS-12. However, the menthol binding site identified by the authors from their docking model is distinct from where the menthol-like moiety in WS-12 is positioned in the cryo-EM structure; the menthol molecule is not capped by Y745 but is located above the WS-12 binding site and more importantly, Y745 and Y1005 do not form prominent interactions with the menthol molecule, which is inconsistent with the previously reported functional studies by Bandell et al. Therefore, the binding configuration of menthol depicted in this study is questionable.

We fully agree with the reviewer and the previous reports that Y745 and Y1005 are important for menthol activation. In fact, our own data confirmed the Bandell's report that the Y745H mutation was barely activated by menthol (Fig. 1a). To measure EC50 and Pomax from Y745 site, we employed the mutation Y745F (Fig. 2g), which in Bandell's paper it has been reported to be still functional and menthol activated (Bandell *et al.*, Nature Chemical Biology, 2006, Fig. 3c). With the Y745F mutant, we measured its coupling energy with the hydroxyl group of menthol and observed a small value (Fig. 2g). Therefore, what our data suggest is that Y745 and the hydroxyl

group of menthol does not specifically interact with each other, it does not mean that the Y745 residue is not important for menthol binding.

From the structure perspective, Yin's cryo-EM structures have revealed that in the apo state, the sidechain of Y745 points toward R842 and occupies the putative ligand binding pocket (Fig. 1d). Our docking showed that when menthol is bound, the sidechain of Y745 swings to open the binding pocket, so that Y745 will sandwich the menthol between its sidechain and R842 (Fig. 1e), just like Yin's observation of the menthol analog WS-12. In our docking model, Y745 prominently contribute to menthol binding through its VDW interactions with the menthol molecule (Fig. 1h, top right). In this way, Y745 contributes significantly to menthol binding likely by its bulky sidechain, as both Bandell's study and our own data show that when the benzene group is kept in Y745F mutant, the menthol still activates the channel, but once the benzene group is removed in Y745H mutant the menthol activation is virtually abolished.

As for the Y1005 site, Bandell's work shows that on Y1005F mutant, the menthol activation was reduced as compared to the WT but not abolished (Bandell *et al.*, Nature Chemical Biology, 2006, Fig. 1b). We also observed that in patch-clamp recordings, the EC₅₀ of menthol activation was increased from $94.5 \pm 4.5 \mu\text{M}$ of WT to $498.4 \pm 8.7 \mu\text{M}$ of Y1005F, suggesting the menthol activation was indeed weakened on the Y1005F mutant. What our thermodynamic mutant cycle analysis suggest is that Y1005 site does not specifically interact with the hydroxyl group and the isopropyl group of menthol molecule. Therefore, we speculate that Y1005F mutation reduces menthol activation through an allosteric mechanism.

To further illustrate our docking model is compatible with previous cryo-EM structures, we have aligned our docking model with the structure of WS-12 bound state (PDB ID: 6NR2) (Fig. S8). We observed that both the menthol and the menthol-like moiety of WS-12 (Fig. S8e, dashed box in red) molecules are sandwiched between Y745 and R842 but in a slightly different way: for menthol, the sidechain of Y745 points downward, while the sidechain of R842 points upward (Fig. S8d); for WS-12, the sidechain of Y745 points upward, while the sidechain of R842 points downward (Fig. S8e). When these two binding configurations are superimposed (Fig. S8f), we observed that the menthol molecule binds upward to the menthol-like moiety of WS-12.

We think that such a binding configuration of menthol is reasonable. From the perspective of chemical structures of menthol and WS-12, it is obvious that WS-12 is nearly twice as large as menthol (Fig. S8a and S8b). While the menthol-like moiety of WS-12 is sandwiched between Y745 and R842, its benzene ring moiety (Fig. S8e, dashed box in blue) points downward to the entrance of ligand bind pocket and the TRP domain beneath. The benzene ring moiety of WS-12 likely interacts with Y1005 on the TRP domain as revealed by Yin's cryo-EM structure, so when this benzene ring

moiety is removed as in the menthol molecule, the menthol can no longer directly interact with residues on the TRP domain. Then menthol binds to a deeper position within the ligand binding pocket formed by the S1-S4 domain.

To test whether the menthol molecule can stably bind in such a configuration suggested by our docking, we perform molecular dynamic simulations. It is known that if a small molecule ligand binds weakly to its protein receptor, most likely it will dissociate during the first tens of nanoseconds during molecular dynamic simulation. Starting from our docking model with explicit water and lipid molecules, we observed that menthol binds stably to the vicinity of R842 (Fig. 4h; Supplementary Movie 1) during the 378 ns simulation time, which supports that our menthol binding configuration is reliable.

2) The menthol-induced conformational changes described in this study are contradictory to the agonist-induced structural rearrangements shown in the recently published TRPM8 complex structures. By comparing cryo-EM structures of TRPM8 in the apo and in the agonist-bound states, Yin et al. showed that (1) the binding of the menthol analog WS-12 or the synthetic super-cooling agent icilin triggers an outward rotation of the VSLD away from the pore domain; (2) the binding of the allosterically coupled PIP2 and agonist would induce a bending in the S5 helix. However, in this study, the authors identified that binding of menthol at R842 in S4 induced inward movement of S2 and S3 towards S1 and S4, followed by an outward movement of the S4-S5 linker and opening of the S6 gate. The menthol-induced conformational changes based on the computational modeling are questionable, so the authors cannot suggest coupling between menthol and voltage in TRPM8 gating using these results.

We need to clarify that we suggest the coupling between menthol and voltage activation in TRPM8 first based on our patch-clamp recordings (Fig. S6). We observed that deep beyond -200 mV decreased the current through TRPM8 channel even in the presence of near saturating concentration of menthol (Fig. S6b, dashed box in red). There when the transmembrane voltage was at +80 mV, menthol-induced current (trace in blue) was much larger than that without menthol applied. As the voltage was decreased to the hyperpolarizing range, we observed a clear “hook” in the current trace as the open probability of TRPM8 was decreased by hyperpolarizing voltage even with the increase in voltage driving force and with the agonist menthol. Finally, the current declined to a level similar to that recorded in the absence of menthol (trace in black, dashed box in red), indicating the TRPM8 channel was fully shutdown by voltage itself, even with menthol applied. In contrast, capsaicin activation of TRPV1 was not antagonized by deep hyperpolarization (Fig. S6a), because capsaicin is known to activate the channel without involving voltage sensing apparatus. With these patch-clamp recordings, we suggest that menthol and voltage activation is coupled. Such a coupling is supported by our docking results, because the menthol molecule directly interacts with the gating charge contributing residue R842.

We agree with the reviewer that our suggested conformational rearrangements induced by menthol is not identical to those induced by WS-12 and icilin. We think it is because the chemical structure of menthol is different from those of WS-12 and icilin (Fig. S8a to S8c). Different ligands induce different conformational changes. In fact, even WS-12 and icilin induce different conformational changes as demonstrated by Yin's work (Science, 2018 and 2019). For instance, compared to the unliganded state (PDB ID: 6BPQ) the bending of S5 helix as mentioned by the reviewer, only occurred in the class 1 icilin-bound state (PDB ID: 6NR3). Such a bending is not observed in the class 2 icilin-bound state (PDB ID: 6NR4) or in the WS-12-bound state (PDB ID: 6NR2) (Fig. S7, dashed boxes in red).

Moreover, menthol is much smaller than these two molecules (Fig. S8a to S8c). When WS-12 is bound to pocket in S1-S4 as revealed in Yin's work, its benzene ring moiety points downward to the entrance of ligand binding pocket, which may prevent the inward movements of S2 and S3. Icilin has two benzene ring-like moieties joined by a central dihydropyridine-like moiety, which is even larger than WS-12 and menthol so that it interacts with more residues within the S1-S4 ligand binding pocket. Therefore, we believe that it is reasonable that menthol-induced conformational rearrangements are different from those induced by other agonists.

3) I realized that instead of using the WS-12 and PIP2-bound TRPM8 structure as a template, Xu et al. chose to use the apo TRPM8 structure. It is a well-established fact that TRPM8 requires the membrane lipid PI(4,5)P2 for the channel activation and depletion of PI(4,5)P2 leads to channel desensitization (or rundown). Furthermore, menthol-mediated TRPM8 activation is allosterically modulated by PI(4,5)P2, and the binding of menthol and PI(4,5)P2 is allosterically coupled. The computational studies for menthol docking as well as their open state model did not include PI(4,5)P2 but the computational studies led to an open state, further calling into a question the results from modelling studies.

As pointed out by the reviewer, we have used the apo TRPM8 structure to first model the closed state of our mouse TRPM8 channel. This is reasonable because the closed state we modeled is indeed an apo state without any ligand bound.

For docking of the menthol molecule into its binding pocket formed by the S1-S4 domain, we observed that as revealed in Yin's work (Science 2018 and 2019), structures of this binding pocket in either the apo state (PDB ID: 6BPQ), the class 1 icilin-PIP2-Ca²⁺ bound state (PDB ID: 6NR3), the class 2 icilin-PIP2-Ca²⁺ bound state (PDB ID: 6NR4) and the WS-12-PIP2 bound state (PDB ID: 6NR2) are virtually identical with a RMSD less than 0.9 Å (Fig. S7d). The slight variations in the structure of ligand binding pocket as observed in Yin's work are induced by either icilin or WS-12, but not by menthol itself. So these structures of icilin bound or

WS-12 bound S1-S4 domain are less suitable to dock menthol molecule. Therefore, it is reasonable to dock menthol molecule to the apo state.

For modeling of the menthol-induced open state of TRPM8, we resorted to the constraints derived from ANAP imaging experiments, which were conducted in live cells at room temperature. So besides the PI(4,5)P2 mentioned by the reviewer, live cells contain other modulators (even unknown to date) of TRPM8 channel at their physiological concentrations and conditions. In this sense, changes in ANAP fluorescence emission induced by menthol activation have already implicitly integrated modulatory effects of PI(4,5)P2 and beyond on the TRPM8 channels, making our ANAP shifts and constraints derived more physiologically relevant.

As pointed out by the reviewer in Major Concern (7), ANAP emission does not translate to the distance changes. However, changes in ANAP emission indicate changes in SASA, which can be calculated for the sidechain of a given residue from the 3D structure model. When a right shift in ANAP emission was observed at a specific residue, in modeling of the open state we required that the SASA of this residue must increase by at least 10 \AA^2 compared to the closed state (see the Methods for detail). This is based on our previous study on capsaicin-induced open state of TRPV1 channel, where we demonstrated that ANAP emission shifts measured from live cells can be used as SASA constraints to computationally derive the open state of TRPV1 (Yang et al., Nature Communications, 2018).

In our initial submission we have employed ANAP information from sites 824, 875, 920, 922, 939, 958 and 966 (Fig. 5 and Fig. 6) to model the menthol-induced open state. From a modeling technical point of view, in revision of this manuscript we have performed more ANAP imaging experiments on the S1-S3 domains as suggested by Reviewer #1. We observed that the ANAP emission shifts measured at sites 774 and 791, which were not included as constraints in the initial model-building process, also agree with the SASA changes calculated from our closed and open state models (Fig. S5d, data points in purple). In this way, our models in both states have been again validated by our new ANAP imaging experiments. Starting from the apo and closed state model, we employed the more physiologically relevant ANAP constraints to model the menthol-induced open state, so our models are both technically and physiologically reliable.

4) I realized that the molecular docking approach is biased, because the authors initially placed the menthol molecule roughly in the center of the cavity defined by S1 to S4, rather than using the full channel and starting from a random site.

We need to clarify that the purpose of docking is not to find out potential binding sites of a molecule, but to suggest potential atomic details of the interaction between a molecule and its protein receptor. For a docking experiment to make scientific sense,

certain knowledge of the ligand binding site must be known in advance. Specifically, for the binding of menthol to TRPM8 channel, as the reviewer has pointed out in Question (1), Bandell's work showed that mutations within the pocket formed by S1-S4 domain largely affected menthol activation. Yin's cryo-EM structures reveal that menthol analog WS-12 and icilin bind to the same pocket formed by S1-S4 domain. More importantly, Voets' work (Voets, T. *et al.*, Nature Chemical Biology, 2006) demonstrated the R842A mutation significantly decreases the binding of radio-actively labelled menthol molecules. Therefore, we believe that it is reasonable for us to hypothesize that the menthol binding pocket locates inside the cavity formed between the S1-S4 domain near the R842 residue. We performed docking and patch-clamp recordings to test this hypothesis, and we found that menthol indeed binds to this pocket as its hydroxyl "hand" specifically interact with R842, and its isopropyl "legs" specifically interact with L843 and I846.

5) I realized that authors did omit the comparison of their structural and mechanistic models with recent complex structures of TRPM8 by the Lee group as well as previous functional data by the Patapoutian group. The authors should make comparisons and discuss the differences.

We have added the comparison of our docking model and cryo-EM structures of TRPM8 by the Lee group in Fig. S8. Like our responses to the reviewer's previous questions, we have added discussion of such comparison with cryo-EM structures and the mutagenesis work from the Patapoutian group in the main text.

During the revision of our manuscript, the Julius and Cheng groups have also determined the cryo-EM structures of TRPM8 channel in the apo state, AMTB-bound closed state, TC-I 2014-bound closed state and Ca²⁺-bound desensitized state (PDB ID: 6O6A, 6O6R, 6O72 and 6O77, respectively). We have also compared our docking model with these structures (Fig. S9). We observed that protein structures of the ligand binding domain formed by the S1-S4 are again almost identical in different states, as Yin's work has shown (Fig. S7d). Moreover, the antagonists TC-I 2014 and AMTB all bind near the entrance of this binding pocket, preventing the binding of menthol deep inside this pocket.

6) Based on the docking model, to validate specific interactions between residues in the TRPM8 VSLD cavity and menthol functional groups, the authors performed thermodynamic mutant cycle analysis between VSLD mutants and menthol analogs. The caveat for this approach is the use of double-mutant cycle analysis between protein and (a very) small molecule. Double mutant cycle analysis is a useful tool when applied to protein-protein interactions, but not so much for the small molecule-protein interaction because removing or mutating a functional moiety in the small molecule often leads to a completely different interaction profile with protein. In particular, menthol exhibits a relatively small chemical structure. The authors removed its hydroxyl or isopropyl group and measured the apparent binding affinity

to the wildtype or mutant channel. Binding of these modified menthol analogs might bind nonspecifically to a location elsewhere in TRPM8. In this case using double-mutant cycle analysis to probe interactions between menthol and residues in VSLD lacks legitimacy. Although the Zheng group used the same approach to study the mechanisms of capsaicin binding and TRPV1 channel activation, the double-mutant cycle analyses between capsaicin and TRPV1 channel were systematically designed. Capsaicin is larger than menthol and contains amphipathic nature. Furthermore, key functional groups other than aliphatic tail in capsaicin were systematically replaced but not removed, which preserve its interaction profile with TRPV1.

We agree with the reviewer that for a small molecule like menthol, removing a group from its chemical structure may largely affect its binding. For the isopropyl “leg” in the menthol, in the initial submission we removed this functional group in double mutant cycle analysis, which may cause unwanted effects as the reviewer pointed out. Therefore, in revision of our manuscript, we have employed another menthol analog isopulegol (Fig. 4). Compared to menthol, isopulegol has the same number and type of atoms, but instead of the isopropyl group of menthol, isopulegol has the methylethenyl group containing a double bond between carbon atoms instead of a single bond. By using this analog, we induced subtle changes to menthol chemical structure without removing a functional group. With isopulegol, we performed double-mutant cycle analysis and again observed large coupling energies between the “legs” of menthol with L843 and I846 of TRPM8 channel, which consolidate our previous observations.

For the hydroxyl “hand” of the menthol molecule, we first observed that if the hydroxyl group is removed as in the analog p-menthane, p-menthane was not able to activate TRPM8 channel (Fig. 2c), preventing us from using this analog for double-mutant cycle analysis. Nonetheless, our observation that p-menthane failed to open TRPM8 demonstrates the importance of the hydroxyl group in menthol. So to subtly modify the hydroxyl group, we employed another analog menthone, where the hydroxyl group is not removed but replaced by a carbonyl group (Fig. 2d). With menthone, we observed a large coupling energy between the hydroxyl group and the R842 residue. Therefore, our double-mutant cycle analysis with three menthol analogs (menthone, isopulegol and 3-methylcyclohexanol) clearly suggested that the hydroxyl “hand” interacts with R842, while the isopropyl group points downward to L843 and I846. Such a binding configuration is also stable in molecular dynamic simulations (Fig. 4h; Supplementary Movie 1), supporting the reliability of our docking results.

7) Authors computationally determined TRPM8 structure in the closed state and employed ANAP imaging to reveal residues that showed conformational changes induced by menthol binding. The ANAP fluorescence data were incorporated for computational modeling of the menthol-induced open state, where the changes in the solvent accessible surface area (SASA) of the side chain were imposed as constraints.

However, ANAP only provides information on accessibility change in a qualitative manner, but it does not provide information (e.g. distance) that can be used as constraints for modeling. It is unclear how ANAP data can be incorporated into modeling an open state. For example, the large ANAP signal cannot be interpreted as large conformational changes and can be interpreted in multiple ways. As far as I know there are no systematic studies to correlate ANAP signals for structural changes. Residues showing ANAP signal shifts are located at the ion permeation pathway-lining region and thus it is difficult to draw any meaningful mechanistic insights.

As discussed in response to the Question (3), we agree with the reviewer that ANAP emission shift does not translate to the distance changes. However, changes in ANAP emission indicate changes in SASA, which can be calculated for the sidechain of a given residue from the 3D structure model and used as constraints in structural modeling. When a right shift in ANAP emission was observed at a specific residue, in modeling of the open state we required that the SASA of this residue must increase by at least 10 \AA^2 compared to the closed state (see the Methods for detail), which is based on our previous studies on the correlation between ANAP signals and SASA changes.

In our study of the capsaicin activation of TRPV1 channel, we observed that capsaicin-induced ANAP changes and SASA changes were indeed correlated, which allowed us to use such information as constraints to model the capsaicin-induced open state of TRPV1 (Yang et al., Nature Communications, 2018). In addition, when we studied the heat desensitization of TRPV1 channel, we also observed the correlated changes in ANAP signals and SASA and used SASA changes to computationally model the heat-desensitized state of TRPV1 (Luo et al., Nature Communications, 2019). Therefore, our way of using ANAP emission to constrain computational modeling has been validated by previous studies, as well as additional ANAP experiments during the revision of this manuscript discussed below.

As also suggested by Reviewer 1, we have performed more ANAP imaging experiments for residues on the S1-S3 domains. There we found eight sites (734, 774, 787, 789, 791, 795, 808 and 817) that were tolerant to ANAP incorporation (Fig. 5f). Among these eight sites, ANAP incorporated at site 774 of the S2 and 791 of the S3 exhibited significant right shifts larger than 2 nm in emission spectra during menthol activation of the channel. Such shifts in ANAP emission indicate that the S2 and S2-S3 linker undergo conformational changes during menthol activation.

More importantly, as discussed in response to Reviewer 1's concerns, the right shifts in ANAP emission measured at additional sites 774 and 791 serve as validating data for our computational modeling of menthol-activated open state. Our current model was built with ANAP constraints at sites 824, 875, 920, 922, 939, 958 and 966 (Fig. 5 and Fig. 6). We observed that the SASA changes calculated from our models at sites

774 and 791, which were not included as constraints in the model-building process, also agree with the ANAP shifts measured experimentally (Fig. S5d, data points in purple). Therefore, we believe that our additional ANAP measurements have not only validated our computational models, but also revealed conformational changes in TRPM8 induced by menthol binding.

Minor comments

1) In methods, line 420 and 423, the authors wrote “dock capsaicin to TRPM8”. It should be “menthol”.

Thank the reviewer and we have fixed this typo.

2) Labels for residues and helices are missing in Figure 1f-g.

We have added labels for residues and helices in Fig. 1f and 1g. We have also added a panel illustrating the topology of a TRPM8 subunit (Fig. S1c).

3) In Fig. S1, the single-channel current traces are too noisy and appear to have multiple subconductance.

We have optimized our single-channel recordings setup to reduce the noise, so we have updated the current traces in Fig. S1. Moreover, we performed single-channel recordings as in Fig. S1 to show that menthol is a partial agonist to TRPM8 channel, so that the open probability cannot reach unity by menthol activation. Indeed, the single-channel recordings and the histogram of single-channel current (Fig. S1a and S1b) clearly demonstrated this point.

Reviewer #3 (Remarks to the Author):

Summary of paper:

Experiments and computations are used to suggest a mechanism of how menthol binds and activates its receptor: the TRPM8 ion channel. The computational methodology includes structural modeling and docking, as well as elastic network model analysis of the equilibrium dynamics, and the experimental techniques include mostly electro physiology, double-mutant cycles, and using unnatural fluorescence amino acid.

Opinion:

The problem is interesting and important, and the approach suitable. The interplay between computation and experiment helps to decipher the activation mechanism in great details. The first part, with menthol docking is very convincing; the stereochemistry of the binding pose makes sense. The second part, where elastic network analysis is used to suggest the consequent conformational changes in the channel is a bit less convincing. Overall, it is a very good manuscript that is suitable for publication after relatively minor changes.

Major issues:

1. A bit more work is needed to further support the suggested conformational changes in TRPM8 upon menthol binding. Here are two suggestions in this direction (but there could be better ideas): (a) the elastic network model could reveal hinge residues that are important for the conformational changes. These could be examined in mutagenesis. (b) Evolutionary conservation analysis, e.g., using ConSurf (<http://consurf.tau.ac.il>) could be used to highlight key amino acids that are important for the mechanism. It would be helpful to correlate these with the observations from the dynamics.

We appreciate the reviewer's suggestions to further test the conformational changes in menthol activation. As also suggested by reviewer 1, we have first performed the rate-equilibrium linear free-energy relationship analysis (Φ analysis) with single-channel recordings to corroborate the elastic network modeling. As described in response to Reviewer 1's comments, we made 47 point mutations on TRPM8 and measured Φ values from four different sites on TRPM8 (Fig. 7). Our experimentally measured Φ values correlate well with the Φ values calculated from the iENM (Fig. 8), supporting our proposed temporal sequence of channel-activation events upon menthol binding.

Furthermore, we performed evolutionary conservation analysis with the ConSurf server. We observed that as pointed out by the reviewer, residues critical for menthol activation are indeed highly conserved in evolution (Fig. S2). For instance, in the ligand binding pocket, residues that directly interact with the menthol molecule like

R842, L843 and I846 are all well conserved. R842 interacts with D802 in the apo state as revealed by cryo-EM structures, so D802 is also highly conserved. We have introduced point mutations at these sites and found large impact on menthol activation (Fig. 2).

2. Modelling the selectivity filter in a closed and open states: Apparently, the authors have an entire paper about this modelling project but it is still in press. From what they describe here, they used Rosetta de novo modelling. Since loop modelling is tricky, it is worthwhile to try some other methods and compare the results. For example, the TRPM2 channel has a 45% sequence identity to the TRPM8 channel. TRPM2 structure was solved in both the apo-closed and Ca²⁺-bound open states (6MIX and 6MJ2 respectively). It seems that TRPM2 structure includes the loop regions that are missing here. This might be a better starting point for modelling both the closed and open states compared to de novo modelling. In addition, it would be helpful to run ConSurf and check that the amino acids facing the filter are the most highly conserved, as they should. The ConSurf analysis could also help select the most suitable model of the loops among various decoys.

While this manuscript is in revision, our paper on the modeling of the selectivity filter of TRPM8 has been published (PMID: 31102353). Though as the reviewer pointed out, several structures of TRPM2 channel have been published, the relatively low sequence identity (45%) between TRPM8 and TRPM2 prompted us to perform *de novo* modelling with the experimentally derived constraints from our ANAP imaging experiments. Previously we have employed such an ANAP-constrained modeling approach to derive the capsaicin-induced open state of TRPV1 channel (Yang *et al.*, Nature Communications, 2018). In revision of this manuscript, we performed additional ANAP imaging experiments for residues in the S1-S3 domains. Our observed right shifts in ANAP emission at sites 774 and 791, which were not included as constraints in our previous model-building process, are also consistent with the prediction of SASA changes from our current TRPM8 models in the closed and open states (Fig. S5). In this sense, our models are validated again by these new data. Therefore, we prefer using the *de novo* modeling approach for the selectivity filter.

We performed the ConSurf analysis and observed that in the pore region of TRPM8, the S5, pore helix, selectivity filter and S6 are conserved, while the turret between the selectivity filter and S6 is variable (Fig. S2). The turret is known to be under positive selection pressure so that its sequence is expected to be variable. As predicted by the reviewer, we observed that the conserved residues P916 indeed locates facing the selectivity filter (Fig. S2). When we performed Φ analysis with point mutations and single-channel recordings at this site, we observed that the Φ value there is the smallest among tested sites, indicating the upper gate near the selectivity filter moves later than ligand binding and lower gate at the S6 bundle crossing. Our iENM analysis also showed that P916 is one of the late moving residues, indicating a critical role of this site in the gating of TRPM8. Therefore, ConSurf analysis consolidated both our

models and functional studies.

Minor issues:

1. Paragraph starting in line 57: A figure/supplementary figure showing the different domains and the nomenclature of the helices could help readers that are not familiar with this channel.

We have added a diagram of the topology of TRPM8 channel in Fig. S1c as suggested by the reviewer.

2. Line 144: what is Fig. 1s?

It should be “Fig. 1c”. We have fixed this typo in the revised manuscript.

3. Line 420 and line 423: Presumably menthol rather than capsaicin.

We thank the reviewer and have fixed this typo.

4. Figure 1 f and g, amino acids numbering is missing.

We have added the amino acid numbering in Fig. 1f and 1g as suggested by the reviewer.

5. General comments concerning the figures: Jamming 8 different panels into one figure can be confusing.

We appreciate the reviewer’s suggestion and have made new figures and rearranged the panels in the revised manuscript.

6. The manuscript reads very well overall, but a few sentences could be further edited.

To improve our manuscript, we have had the manuscript language proofed by native English speakers.

Peer Review File

Reviewers' comments second round:

Reviewers' comments:

Reviewer #1 (Remarks to the Author):

Authors performed new experiments and carefully reedited the manuscript. All concerns which I raised are now addressed satisfactorily in this version. Congratulations on a completion of a nice piece of work on gating mechanisms of TRPM8 channel!

I have only a few editorial comments as below.

1. Line 177; Fig. S2d is probably Fig. S3d.
2. Line 612 Cl²⁻ should be Cl⁻.

Reviewer #2 (Remarks to the Author):

The authors responded to some of my previous comments by performing additional experiments and the revision has been improved. While this reviewer appreciates the authors' efforts on the additional experiments, there are still issues that need to be addressed. Importantly, I realized that the modeling studies were conducted with a fragment of the channel by omitting important structural subdomains, which could be the reason for the apparent discrepancy between the results from functional and modelling studies.

Major comments:

1) The location of menthol in the TRPM8 model is different from those of other TRPM8 agonists and antagonists from cryo-EM structures. The authors performed additional double mutant cycle studies and found that two residues (L843 and I846) on S4 are energetically coupled to menthol. While this reviewer appreciates the efforts by the authors on the additional experiments, the mutant cycle results are contradictory to their models. For example, Fig. 3a suggested that I846 and L843, two residues that are energetically coupled to menthol, are located too far away from menthol to make direct interactions (authors did not provide any distances between in Fig. 3a). Indeed, their model indicates the distances from I846 and L843 to menthol are ~ 4.8 Å. In addition, in their model, while the distance of R842 to menthol is ~ 3 Å, that of D802 (another residue that is energetically important for menthol binding in this study) is ~ 5.4 Å, which is again too far to make direct interaction with menthol. The authors' assumption is that residues with coupling energy above 1.5 kT form direct interactions with menthol. However, the thermodynamic cycle data, although nicely done, is contradictory to the model. In fact, if the menthol is positioned similar to the agonists or antagonists in the cryo-EM structures, the interaction distance between the leg position of the menthol and I846 will be closer. Therefore, the mutant cycle data does not support the menthol binding site from the modeling study. It is either that the high coupling energy between menthol and I846, L843, and D802 indicates allosteric coupling, not direct interaction or the proposed menthol site in the modelling is incorrect. The authors need to perform direct binding studies (e.g. radiolabeled menthol) for both wildtype and I846V or L843A TRPM8 to test whether menthol interacts with these two residues directly. With the results from the binding studies, the authors can dissect these two possibilities.

2) After looking at the models (PDB files) provided in the source data, this reviewer realized that the modeling for menthol binding was done with only the transmembrane segments (S1-S4) and without the TRP domain. Importantly, although the method was not described in detail, the modeling studies for closed and open states were done using the transmembrane tetramer excluding the TRP domain as well as the cytosolic domain (including MHR1-MHR4). Among all the omitted subdomains, the TRP domain is especially essential to TRPM8 function. It serves as the binding site for menthol, and is also key to the menthol-dependent channel gating. Therefore,

omitting the TRP domain in the modeling studies could have resulted in the discrepancy between the experimental and modeling studies. Also, because the TRP domain fills the large void in the cytoplasmic side of the S1-S4 subdomain, its absence in the modeling studies could have led to the motion in the proposed open state, which is physically not feasible (it appears to me that the S4-S5 linker moves to where the TRP domain should be located). The authors should perform control modelling studies or perform modelling studies using the intact channel structure. For example, using the fragment (the S1-S4 subdomain without the TRP domain) that they used for the menthol binding studies, they should perform the modelling studies for the published agonists or antagonists and see if they bind to the positions revealed in the cryo-EM structures. For the open state model, at minimum, the authors should perform the modelling studies in the presence of the TRP domain and the menthol (I found no menthol in the open state model).

3) This reviewer previously raised the point that the absence of PIP2 in their modeling studies as a caveat, as PIP2 is essential to TRPM8 function. The authors responded that they resorted to constraints derived from in vivo fluorescence (ANAP) studies for modelling. In the abstract, the authors stated that the information from imaging experiments was integrated into structural model construction. However, in the newly incorporated text (lines 299-303), the authors commented that the SASA changes from the ANAP studies were not included in modeling. These are contradictory to each other, so the authors need to clarify how exactly the model was built throughout the text. Because ANAP studies do not generate distance constraints, they cannot be used for generating structural models. After the model was built, the SASA changes from the ANAP studies can be used to see if the model is consistent with the data.

4) The authors should refrain from using the term upper gate throughout the text, and the notion that the selectivity acts as a gate. There is no experimental data suggesting whether the selectivity filter of TRPM8 acts as an activation gate. In line 245-252 and Fig. 5c, the authors describe their model in which the selectivity filter is in closed conformation. However, in comparison to their computational model, the outer pore region, including the pore helix, pore loop, and the selectivity filter of the published calcium-bound TRPM8 structure by the Julius and Cheng groups show drastic distinct conformations (Fig. S10). The authors' explanation of "structural variability" is not a good way to justify the difference between the computational model and the cryo-EM structures. The Swartz lab's recent data suggest that even the selectivity filter of TRPV1-TRPV3 does not act as a gate but do suggest that there are some conformational changes around the selectivity filter, thus the SASA changes observed from the ANAP studies cannot be used as an evidence for the selectivity filter as a gate in TRPM8. To maintain the notion of the upper gate, the authors must perform additional experiments to show state dependent accessibility change at the selectivity filter.

Minor comments:

1) Please provide all the distances between menthol and the residues shown in Figs. 2a and 3a, and compare their distances with the coupling energy values shown in Figs. 2g and 3g.

2) The two references that the authors chose to support selectivity filter as a gate are not relevant, so I suggest they should be removed. The reference for Shaker potassium channel (ref. #25) is about selectivity change from the mutant channel and not about the gating. The reference for the MthK studies is about voltage-dependent block, not about calcium dependent selectivity filter conformation changes, thus it is irrelevant. Ref. 25 is about the mutant Shaker channel which exhibits two sub-conductance level with different ion selectivity. However, this paper does not address whether in Shaker channel the selectivity filter acts as an activation gate. Actually, it is established that only the bundle crossing region on S6 acts as the activation gate in Shaker by many laboratories (for example, Gary Yellen's well-established accessibility studies).

3) In line 427-434, the authors commented that the orientation of D918 facing the central ion permeation pathway the same as that of D907 and D908 in the cryo-EM structure; in contrast, the aspartic acids are much farther apart in the reported structure (Fig. S10). The authors need to address more about the discrepancy between the model and the available structural data than simply "structural variability" in the selectivity region (line 434).

4) There is not description about how the authors obtained the open probability value (P_o) in

Figure 2 and 3. Do any mutants reveal basal activity such that P_o is larger than 0 in the absence of ligand?

5) In the abstract, "structures" (line 38) should be replaced with "models".

6) The residue numbering in both open and close models, which only show TMD regions, starts from #1-#245. This is very misleading for readers trying to locate important residues and compare the selectivity filter and S6 gate with published cryo-EM structure. If the models were made for mouse TRPM8, please change to the corresponding mouse TRPM8 residue numbers.

7) There are still typos. Line 132 "...two residues R842 and D802 that may from a hydrogen bond": from should be replaced with form.

8) Line 340 "As...are in serial (Fig. 4C)"; do you mean "in series"?

Reviewer #3 (Remarks to the Author):

The authors have addressed well all the issues raised and the manuscript is now perfectly suitable for publication.

Nir Ben-Tal

Reviewers' comments:

Reviewer #1 (Remarks to the Author):

Authors performed new experiments and carefully reedited the manuscript. All concerns which I raised are now addressed satisfactorily in this version. Congratulations on a completion of a nice piece of work on gating mechanisms of TRPM8 channel!

Response: We appreciate the reviewer's constructive comments to improve our manuscript.

I have only a few editorial comments as below.

1. Line 177; Fig. S2d is probably Fig. S3d.

Response: Thank the reviewer and we have fixed this typo.

2. Line 612 Cl2- should be Cl-.

Response: Thank the reviewer and we have fixed this typo.

Reviewer #2 (Remarks to the Author):

The authors responded to some of my previous comments by performing additional experiments and the revision has been improved. While this reviewer appreciates the authors' efforts on the additional experiments, there are still issues that need to be addressed. Importantly, I realized that the modeling studies were conducted with a fragment of the channel by omitting important structural subdomains, which could be the reason for the apparent discrepancy between the results from functional and modelling studies.

Major comments:

1) The location of menthol in the TRPM8 model is different from those of other TRPM8 agonists and antagonists from cryo-EM structures. The authors performed additional double mutant cycle studies and found that two residues (L843 and I846) on S4 are energetically coupled to menthol. While this reviewer appreciates the efforts by the authors on the additional experiments, the mutant cycle results are contradictory to their models. For example, Fig. 3a suggested that I846 and L843, two residues that are energetically coupled to menthol, are located too far away from menthol to make direct interactions (authors did not provide any distances between in Fig. 3a). Indeed, their model indicates the distances from I846 and L843 to menthol are ~ 4.8 Å. In addition, in their model, while the distance of R842 to menthol is ~ 3 Å, that of D802 (another residue that is energetically important for menthol binding in this study) is ~ 5.4 Å, which is again too far to make direct interaction with menthol. The authors' assumption is that residues with coupling energy above 1.5 kT form direct interactions with menthol. However, the thermodynamic cycle data, although nicely done, is contradictory to the model. In fact, if the menthol is positioned similar to the agonists or antagonists in the cryo-EM structures, the interaction distance between the leg position of the menthol and I846 will

be closer. Therefore, the mutant cycle data does not support the menthol binding site from the modeling study. It is either that the high coupling energy between menthol and I846, L843, and D802 indicates allosteric coupling, not direct interaction or the proposed menthol site in the modelling is incorrect. The authors need to perform direct binding studies (e.g. radiolabeled menthol) for both wildtype and I846V or L843A TRPM8 to test whether menthol interacts with these two residues directly. With the results from the binding studies, the authors can dissect these two possibilities.

Response: We thank the reviewer for pointing out this apparent inconsistency between the menthol docking model to TRPM8 channel in the closed state and our thermodynamic mutant cycle analysis. As TRPM8 channel is allosterically activated by menthol, our thermodynamic mutant cycle analysis with patch-clamp recordings reflected the dynamic interaction of menthol molecule and the channel in the activated state. Therefore, in this revised manuscript we remodeled TRPM8 channel in the menthol activated state and docked menthol into the channel in the activated state, which was then in full agreement with our patch-clamp recordings. We want to emphasize that our docking study served as a guide for our patch-clamp recordings in live cells, so that we made our conclusions on the binding configuration of menthol molecule based on our thermodynamic mutant cycle analysis but not purely on the docking, because patch-clamp recordings and thermodynamic mutant cycle analysis were performed in live cells and better reflected the menthol binding under physiological conditions.

As the reviewer suggested in the second major comments, we remodeled TRPM8 channel in the menthol activated state in the presence of the TRP domain and docked menthol into the activated state model (Fig. 6a to 6c). To first probe conformational changes in the TRP domain during menthol activation, we incorporated ANAP into nine residue sites on the TRP domain, among which there were three ANAP-incorporated mutants remained to be menthol activated (Fig. 5f). We observed no significant shifts in ANAP emission during menthol activation (Fig. 5f), suggesting that most likely conformational rearrangements induced by menthol at the TRP domain are minor. Therefore, as the reviewer suggested that menthol and its analog WS-12 may share similar binding pocket and configuration, we constructed the model of the menthol activated state using the conformation of the S1-S4 domain and TRP domain from the cryo-EM structure with WS-12 bounded (PDB ID: 6NR2).

When we docked the menthol molecule into the activated state model, we found that the top 10 models with the largest binding energy were well converged (Fig. S6a). Again, the hydroxyl group of menthol formed a hydrogen bond with the sidechain of R842 (O1 atom of menthol to NE atom of R842: 3.17 Å; O1 atom of menthol to HE atom of R842: 2.16 Å) (Fig. 6a and Fig. S6b). More importantly, the carbon atoms in the isopropyl group of menthol are about 3.53 and 3.66 Å from the sidechains of I846 and L843 (C9 atom of menthol to CD1 atom of I846: 3.53 Å; C8 atom of menthol to CB atom of L843: 3.66 Å), respectively (Fig. 6a). So these distances between the docked menthol molecule and TRPM8 channel are fully consistent with our thermodynamic mutant cycle analysis. Moreover, the binding configuration of menthol to the activated state was similar to our menthol docking to the apo state (Fig. 6b, menthol colored in cyan and tan, respectively), as the hydroxyl hand of menthol hydrogen bonds with R842 and the isopropyl legs stand above L843 and I846. Furthermore, we observed that the binding energy of menthol to the activated state was significantly larger than that of to the apo state (Fig. 6c), the convergence of top 10 menthol docking models with the largest binding energy in the activated state was also better than that in the apo state (Fig. S6a and Fig. S3a, respectively). Therefore, we believe that the docking model of menthol to the activated state (Fig. 6a) is in several ways better than our previous docking model of menthol to the apo state. We regard our previous docking model of

menthol to the apo state as a starting point to guide our electrophysiology studies in live cells, because though this model was not perfectly matched with our analysis later, it offered important clues such as the interaction with R842 for us to test in wet-lab experiments. Due to the tight regulation of radioactive materials in China and recent outbreak of COVID-19, we were unable to obtain isotope-label menthol as the reviewer suggested. Nevertheless, our revised docking model of menthol is now fully consistent with our thermodynamic mutant cycle analysis. We again fully appreciate the reviewer for this improvement to our manuscript.

2) After looking at the models (PDB files) provided in the source data, this reviewer realized that the modeling for menthol binding was done with only the transmembrane segments (S1-S4) and without the TRP domain. Importantly, although the method was not described in detail, the modeling studies for closed and open states were done using the transmembrane tetramer excluding the TRP domain as well as the cytosolic domain (including MHR1-MHR4). Among all the omitted subdomains, the TRP domain is especially essential to TRPM8 function. It serves as the binding site for menthol, and is also key to the menthol-dependent channel gating. Therefore, omitting the TRP domain in the modeling studies could have resulted in the discrepancy between the experimental and modeling studies. Also, because the TRP domain fills the large void in the cytoplasmic side of the S1-S4 subdomain, its absence in the modeling studies could have led to the motion in the proposed open state, which is physically not feasible (it appears to me that the S4-S5 linker moves to where the TRP domain should be located). The authors should perform control modelling studies or perform modelling studies using the intact channel structure. For example, using the fragment (the S1-S4 subdomain without the TRP domain) that they used for the menthol binding studies, they should perform the modelling studies for the published agonists or antagonists and see if they bind to the positions revealed in the cryo-EM structures. For the open state model, at minimum, the authors should perform the modelling studies in the presence of the TRP domain and the menthol (I found no menthol in the open state model).

Response: We appreciate the reviewer's suggestion to include the TRP domain into the modeling. As we explained in response to the reviewer's previous concern, we have done so by first performing more ANAP imaging experiments with ANAP incorporated into the TRP domain to probe menthol-induced conformational changes. We observed little changes in ANAP emission peak at the three incorporation sites at the TRP domain during menthol activation (Fig. 5f). This observation is not surprising because previous reports showed that a chimeric TRPM8 channel with both its TRP domain and C terminus transplanted from those domains of TRPV1 is still robustly activated by menthol with a similar concentration response like the WT TRPM8 (Brauchi S. *et al.*, Journal of Neuroscience, 2007; Brauchi S. *et al.*, PNAS, 2007). Therefore, as our ANAP imaging experiments provided little constraints on the conformation of the TRP domain, we remodeled the menthol-induced open state using the TRP domain conformation from WS-12 bound state (PDB ID: 6NR2). When we docked the menthol molecule to this new model of TRPM8 channel in the open state, the distances between menthol and important residues are all in agreements with our thermodynamic mutant cycle analysis as explained in our response to the reviewer's first concern.

When we compared this new open state model with the TRPM8 channel structure in the apo state, we observed the conformational changes in the S1-S4 domain were smaller than what we saw with the previous model. The S2 showed certain outward movements in the open state like the icilin-induced

conformational changes reported before (Yin, Y. *et al.*, Science, 2019). S6 bundle crossing gate also showed dilation (Fig. 6d to 6g).

Moreover, we performed extra docking studies to test whether in docking our Rosetta suite can recapture the native conformation of TRPM8 agonists and antagonists revealed in cryo-EM structures. We observed that for the agonist WS-12 and antagonists AMTB and TC-I 2014, our top models with largest binding energy converged well to the ligand binding configuration observed in cryo-EM structures with RMSD values less than 2 Å (Fig. S6d, S6f and S6g). For icilin, our docking recaptured the overall binding orientation of the molecule with deviations in binding location (Fig. S6e). These results demonstrated that our docking protocol and the software are capable of capturing a generally correct binding pose of the TRPM8 ligands. We want to again emphasize that throughout this study, our docking study served as a guide for our patch-clamp recordings in live cells, we made our conclusions on the binding configuration of menthol molecule based on our thermodynamic mutant cycle analysis, which was performed in live cells and better reflected the menthol binding under physiological conditions.

We performed all these docking studies in TRPM8 structure with the TRP domain, so we agree with the reviewer that the intact transmembrane domains and TRP domain are needed, which we remodeled in this revised manuscript. We thank the reviewer for this improvement to our manuscript.

3) This reviewer previously raised the point that the absence of PIP2 in their modeling studies as a caveat, as PIP2 is essential to TRPM8 function. The authors responded that they resorted to constraints derived from in vivo fluorescence (ANAP) studies for modelling. In the abstract, the authors stated that the information from imaging experiments was integrated into structural model construction. However, In the newly incorporated text (lines 299-303), the authors commented that the SASA changes from the ANAP studies were not included in modeling. These are contradictory to each other, so the authors need to clarify how exactly the model was built throughout the text. Because ANAP studies do not generate distance constraints, they cannot be used for generating structural models. After the model was built, the SASA changes from the ANAP studies can be used to see if the model is consistent with the data.

Response: We want to clarify this miss understanding as what we stated in lines 299-303 of the first revised version (NCOMMS-19-13255A) simply means that the extra ANAP imaging results obtained during the first revision at sites 774 and 791 was not included in modeling of the open state. In the original version of this manuscript (NCOMMS-19-13255), the ANAP imaging results at sites A875, D920, T922, L939, P958 and S966 were all used as constraints for modeling of the open state channel pore (Fig. S5d). What we want to emphasize is that, though in the first revised version ANAP information sites 774 and 791 was not used, the open state model built from original version was able to predict correct changes in SASA that was in agreement with ANAP changes at sites 774 and 791 (Fig. S5d). Therefore, ANAP information at sites 774 and 791 serve as a validation of our open state model. From Fig. S5d, we can see that our open state model is consistent with all ANAP imaging results, as data points for all ANAP-incorporated sites located within the blue zones where changes ANAP emission shifts matched with changes in SASA.

Nevertheless, we understand the concern of the reviewer that changes in SASA from the ANAP imaging experiments are not distance constraints, so we have tone down our interpretations from the modeling of open state regarding conformational changes between closed and open states in this second revision of our manuscript.

4) The authors should refrain from using the term upper gate throughout the text, and the notion that the selectivity acts as a gate. There is no experimental data suggesting whether the selectivity filter of TRPM8 acts as an activation gate. In line 245-252 and Fig. 5c, the authors describe their model in which the selectivity filter is in closed conformation. However, in comparison to their computational model, the outer pore region, including the pore helix, pore loop, and the selectivity filter of the published calcium-bound TRPM8 structure by the Julius and Cheng groups show drastic distinct conformations (Fig. S10). The authors' explanation of "structural variability" is not a good way to justify the difference between the computational model and the cryo-EM structures. The Swartz lab's recent data suggest that even the selectivity filter of TRPV1-TRPV3 does not act as a gate but do suggest that there are some conformational changes around the selectivity filter, thus the SASA changes observed from the ANAP studies cannot be used as an evidence for the selectivity filter as a gate in TRPM8. To maintain the notion of the upper gate, the authors must perform additional experiments to show state dependent accessibility change at the selectivity filter.

Response: We agree with the reviewer that currently there is no strong evidence to call the selectivity filter of TRPM8 channel as the "upper gate" for ion permeation, so we have removed the term "upper gate" from the manuscript.

Regarding the conformation of the selectivity filter region in our model, we reasoned that this region must show great structural flexibility as it was not observed in most cryo-EM structures of TRPM8 except the calcium-bound and desensitized state (PDB ID: 6O77). In this sense, the conformation observed in 6O77 represents the structural state of the selectivity filter in a particular condition where presence of calcium ions desensitized the channel. What we observed from the modeling represented another energetically stable state of the selectivity filter in the apo state without any ligand. Therefore, it is not surprising that selectivity filter showed distinct conformations in these two states. We acknowledge that the energetically stable conformation of the selectivity filter may not be the sole conformation this region may adopt, as the selectivity filter is structurally flexible. We have toned down our interpretations regarding conformational changes between the closed and open states.

Minor comments:

1) Please provide all the distances between menthol and the residues shown in Figs. 2a and 3a, and compare their distances with the coupling energy values shown in Figs. 2g and 3g.

Response: We labeled the distances between menthol docked to the closed and open state as the reviewer suggested (Fig. 2a, Fig. 3a and Fig. 6a). We think the menthol docking model in Fig. 2a and 3a served as a guide to our patch-clamp recordings but it does not reflect the binding configuration of menthol in the open state (Fig. 6a), where the distances between menthol molecule and channel residues matched well with our thermodynamic mutant cycle analysis. We added a panel in Fig. S6c to compare the distances versus the corresponding coupling energy values.

2) The two references that the authors chose to support selectivity filter as a gate are not relevant, so I suggest they should be removed. The reference for Shaker potassium channel (ref. #25) is about selectivity change from the mutant channel and not about the gating. The reference for the MthK studies is about voltage-dependent block, not about calcium dependent selectivity filter conformation

changes, thus it is irrelevant. Ref. 25 is about the mutant Shaker channel which exhibits two sub-conductance level with different ion selectivity. However, this paper does not address whether in Shaker channel the selectivity filter acts as an activation gate. Actually, it is established that only the bundle crossing region on S6 acts as the activation gate in Shaker by many laboratories (for example, Gary Yellen's well-established accessibility studies).

Response: We agree with the reviewer that these papers do not directly demonstrate the selectivity filter is a gate in Shaker and BK channels, so we have removed these citations.

3) In line 427-434, the authors commented that the orientation of D918 facing the central ion permeation pathway the same as that of D907 and D908 in the cryo-EM structure; in contrast, the aspartic acids are much farther apart in the reported structure (Fig. S10). The authors need to address more about the discrepancy between the model and the available structural data than simply "structural variability" in the selectivity region (line 434).

Response: similar to our response to the reviewer's major concern 4, we reasoned that the selectivity filter region must show great structural flexibility as it was not observed in most cryo-EM structures of TRPM8 except the calcium-bound and desensitized state (PDB ID: 6O77). The conformation observed in 6O77 represents the structural state of the selectivity filter in a particular condition where presence of calcium ions desensitized the channel. What we observed from the modeling represented another energetically stable state of the selectivity filter in the apo state without any ligand. Therefore, it is not surprising that selectivity filter showed distinct conformations in these two states. We have added more discussion on this point in the revised manuscript.

4) There is not description about how the authors obtained the open probability value (P_o) in Figure 2 and 3. Do any mutants reveal basal activity such that P_o is larger than 0 in the absence of ligand?

Response: We have added more descriptions on the measurements of open probability from noise analysis in the revised method session. We performed single-channel recordings of WT and mutants in the presence of saturating concentration of menthol, which is required in thermodynamic mutant cycle analysis. We did not directly measure the P_o of mutant in the absence of ligand, but we did not observe large spontaneous opening events in any mutants in the absence of agonists. Moreover, based on our own experience on other TRP channels and a previous report (Raddatz N. *et al.*, Journal of Biological Chemistry, 2014), the P_o of TRPM8 approaches to 10^{-5} in the absence of ligand at deep hyperpolarized transmembrane voltage and room temperature. Therefore, the intrinsic P_o of TRPM8 channel must be very small and close to zero.

5) In the abstract, "structures" (line 38) should be replaced with "models".

Response: We replaced "structures" with "models" as the reviewer suggested.

6) The residue numbering in both open and close models, which only show TMD regions, starts from #1-#245. This is very misleading for readers trying to locate important residues and compare the selectivity

filter and S6 gate with published cryo-EM structure. If the models were made for mouse TRPM8, please change to the corresponding mouse TRPM8 residue numbers.

Response: We renumbered the residues in our models based on the mouse TRPM8 channel. We thank the reviewer for pointing it out.

7) There are still typos. Line 132 "...two residues R842 and D802 that may from a hydrogen bond": from should be replaced with form.

Response: Thank the reviewer and we have fixed this typo.

8) Line 340 "As...are in serial (Fig. 4C)"; do you mean "in series"?

Response: Thank the reviewer and we have fixed this typo.

Reviewer #3 (Remarks to the Author):

The authors have addressed well all the issues raised and the manuscript is now perfectly suitable for publication.

Nir Ben-Tal

Response: We appreciate the reviewer's constructive comments to improve our manuscript.

Peer Review File

Reviewers' comments third round:

Reviewer #2 (Remarks to the Author):

The previous revision contained two major issues. They are:

- 1) The docking studies were flawed by using a fragment of the channel lacking an important structural subdomain (the TRP domain).
- 2) The mutant cycle data does not support the menthol binding site from the modeling study. In fact, the thermodynamic mutant cycle result is more consistent with location of the ligand in the ligand-bound cryo-EM structures.

To address these issues, this reviewer asked the authors to 1) redo the docking studies using the full-length channel including the TRP domain, and 2) perform direct binding studies of the involvement of I846 and L843 in menthol binding to corroborate their interpretation of mutant cycle studies.

Major comments

1) In this revision, the authors appear to have done the docking studies using the revised open state model including the TRP domain (but without the cytoplasmic domain). They also did the control docking experiments for the other agonists and antagonists that were reported in the cryo-EM structures. The reported result of new docking studies could replicate cryo-EM structural results to some level. However, it is difficult for this reviewer to evaluate their docking results, as there were no changes in the method for docking and MD simulation as if they did not change their model for docking studies. In lines 566-570 and 643-647 of methods, it appears that only the transmembrane domain without the TRP domain was used for docking and MD simulation. It is unclear to the reviewer whether the TRP domain was included for docking procedure, even though the apo state (PDB 6BPQ) and the open-state model contain the TRP domain. If authors made a mistake not to update the method section in the revision, authors must describe in detail how they performed the new docking studies. In particular, authors must describe the residue range of the channel that was used for each docking experiment as well as the docking procedure (i.e. dock menthol into apo state, into open-state model, and dock four agonists and antagonists to reproduce cryo-EM structures).

With regard to the modeling, please define the regions by residue numbering that were used for the open state model (It is not the full-length channel). Also, it is unclear whether the S1-S4 domain and the pore domain were separately built and combined.

2) Despite this reviewer's request, the authors did not make any effort to perform direct menthol binding studies to show the importance of I846 and L843 in menthol binding (these two residues are important suggested by mutant cycle studies, but too far to interact with menthol in their initial docking model. On the contrary these residues are closer to the menthol moiety in the WS-12 bound structure). Instead, they simply added argument which is incorrect to address the distance between the two residues and menthol (See minor comment #3). Because the mutant cycle data cannot distinguish the locations of menthol between their docking study and cryo-EM studies, authors should acknowledge that with their mutant cycle studies, they cannot derive an unambiguous atomic model of menthol bound state.

3) Because the initial docking as well as MD simulation studies were still done using the model in the absence of the TRP domain. On the other side, they performed menthol docking using the open state model which is more energetically favored as they claimed (line 317-319). Therefore, the initial docking and MD simulation should be removed from the text, and the only the open state model should be described in the manuscript. In addition, the comparison of the menthol binding configuration between docking into apo state versus into the open-state model is over-interpretation and is not meaningful.

Minor comments

1) The authors tend to downplay the importance of the TRP domain in menthol- or ligand-dependent gating of TRPM8 as suggested by little change in their ANAP imaging results. Also, the authors' argument that the TRP domain is not critical for menthol binding is based on the

previously publication by the LaTorre group (Brauchi, J. Neuroscience, 2006) as a supporting evidence. The TRP channel community is well aware of the issues with these studies (e.g. lack of reproducibility and suboptimal experimental design and interpretation) and has considered these studies not reliable. Throughout the manuscript revision, I can't help noticing that the authors tend to cite papers in a biased manner to support their claims (e.g. the previous citations that the authors used to support the selectivity filter gate theory, which authors has now removed upon my request). The functional importance of TRP domain is well supported by multiple solid studies by many groups (Patapoutian, Rohacs, Logothesis). Last, the position of the TRP domain differ upon different agonist binding in the reported structures.

2) The Zagotta group has recently utilized ANAP coupled with TmFRET (Transition metal FRET) studies to obtain distance constraint as well as the directionality of motions associated with gating. With these information, they have successfully derived a model of HCN voltage gating (Dai et al, NSMB 2019). In the current studies by authors, fluorescence change of ANAP alone cannot generate much structural or dynamic information. The author should tone down the interpretation of the ANAP result and should address the limitation of their ANAP experiment.

3) Line 229-236, the authors included statements to address the previous reviewer's comments regarding "L843 and I846 are outside interacting distance with menthol isopropyl group." Authors' reasoning in line 234-236 is incorrect. The distances (4.31 Å between L843, I846 and menthol) measured from the docking model do not reflect any conformational change. If the distances are over 4 Å in the ligand-bound model, it means residues L843 and I846 are unlikely to contribute significant binding with menthol. If the authors want to argue that the distance is averaged over a series of models from the simulation. Then what is the standard deviation of the distance measurement? The bottom line is that the difference between 4.31Å and below 4Å is in fact a big discrepancy.

4) Lines 113-115. This is statement is unnecessary. Multiple ligand-bound structures were published and showed agonists and antagonists binding to the cavity between S1-S4 and TRP domain. The authors based on sequence conservation and claimed more confident, which is redundant.

5) I cannot find supplemental video as well as coordinates for the docked models in this revision. The only coordinates provided is for the newly built open state model.

Reviewer #2 (Remarks to the Author):

The previous revision contained two major issues. They are:

- 1) The docking studies were flawed by using a fragment of the channel lacking an important structural subdomain (the TRP domain).
- 2) The mutant cycle data does not support the menthol binding site from the modeling study. In fact, the thermodynamic mutant cycle result is more consistent with location of the ligand in the ligand-bound cryo-EM structures.

To address these issues, this reviewer asked the authors to 1) redo the docking studies using the full-length channel including the TRP domain, and 2) perform direct binding studies of the involvement of I846 and L843 in menthol binding to corroborate their interpretation of mutant cycle studies.

Major comments

1) In this revision, the authors appear to have done the docking studies using the revised open state model including the TRP domain (but without the cytoplasmic domain). They also did the control docking experiments for the other agonists and antagonists that were reported in the cryo-EM structures. The reported result of new docking studies could replicate cryo-EM structural results to some level. However, it is difficult for this reviewer to evaluate their docking results, as there were no changes in the method for docking and MD simulation as if they did not change their model for docking studies. In lines 566-570 and 643-647 of methods, it appears that only the transmembrane domain without the TRP domain was used for docking and MD simulation. It is unclear to the reviewer whether the TRP domain was included for docking procedure, even though the apo state (PDB 6BPQ) and the open-state model contain the TRP domain. If authors made a mistake not to update the method section in the revision, authors must describe in detail how they performed the new docking studies. In particular, authors must describe the residue range of the channel that was used for each docking experiment as well as the docking procedure (i.e. dock menthol into apo state, into open-state model, and dock four agonists and antagonists to reproduce cryo-EM structures).

We apologize for the missing information in the Method session for docking of menthol to the activated state. We have updated the Method in this revised version (line 575-588). In particular, the TRP domain has been included in the models to perform docking procedure. We docked menthol into the binding pocket formed by S1-S4 domain (residues 734-861) and TRP domain (residues 992-1013) in the potential open state model of TRPM8 built as described in "Molecular modeling" below. The model was then relaxed in membrane environment using the RosettaMembrane application and the model with lowest energy scores were chosen for docking of menthol. For docking of WS-12, icilin, AMTB and TC-I 2014, similar to docking of menthol, the transmembrane domains from beginning of S1 to the end of TRP domain in cryo-EM structure of TRPM8 in the WS-12 bound state (PDB ID: 6NR2), icilin bound state (PDB ID: 6NR3), AMTB bound state (PDB ID: 6O6R) and TC-I 2014 bound state (PDB ID: 6O72) was first relaxed in membrane environment using the RosettaMembrane application, respectively. The models with lowest energy scores were chosen for docking of each ligand.

Since we removed the docking of menthol to the apo state as suggested by the reviewer in the later concerns (major concern #3), we did not add the corresponding information in the revised Methods.

With regard to the modeling, please define the regions by residue numbering that were used for the open state model (It is not the full-length channel). Also, it is unclear whether the S1-S4 domain and the pore domain were separately built and combined.

Residues used for the potential open state model are defined in the revised Methods (line 521-530). As we clearly stated in the Methods (line 548-550 in this revised version and line 625-627 in the previous version), The S1-S4 domain (residues 734-861) and the TRP (residues 992-1013) domain were modeled using the corresponding domains in the cryo-EM structure of WS-12 bound state (PDB ID: 6NR2) as templates for homology modeling in Rosetta. The model of the pore region was built separately (guided by ANAP imaging results) and was combined with the model of S1-S4 domain and the TRP domain, and then further refined by the relax application within the Rosetta suite to ensure compatibility with our ANAP imaging (line 548-550).

Therefore, due to the uncertainties in the way the open state model was built (hence the relative positioning of the S1-S4 domain and TRP domain to the pore domain) and the reviewer's suggestions in the following concerns, we have largely toned down our interpretation of the modeling of the potential open state. In the Results, we have changed the subtitle "Structural mechanisms underlying menthol activation" in the previous version to "Conformational dynamics in menthol activation" in this revised manuscript, where we limited our interpretation of the open state modeling to only "Menthol binding may further lead to widening of the S6 bundle crossing" (line 324). Moreover, we have moved original Fig. 6 on the modeling results to the supplement figures as Fig. S7.

We built the potential open state model of TRPM8 channel to help better explain the results we obtained from patch-clamp recordings and ANAP imaging, so toning down the interpretation of our modeling does not affect the conclusions of our study, because these conclusions on how menthol binds and the wide-spread conformational changes, as well as the conformational dynamics revealed by Φ analysis, are based on our functional and imaging experiments. We hope in future structural biologists can determine the menthol-induced open state of TRPM8 with cryo-EM or X-ray crystallography, which would be very helpful to advance the field of TRPM8 studies (line 409-411).

2) Despite this reviewer's request, the authors did not make any effort to perform direct menthol binding studies to show the importance of I846 and L843 in menthol binding (these two residues are important suggested by mutant cycle studies, but too far to interact with menthol in their initial docking model. On the contrary these residues are closer to the menthol moiety in the WS-12 bound structure). Instead, they simply added argument which is incorrect to address the distance between the two residues and menthol (See minor comment #3). Because the mutant cycle data cannot distinguish the locations of menthol between their docking study and cryo-EM studies, authors should acknowledge that with their mutant cycle studies, they cannot derive an unambiguous atomic model of menthol bound state.

We again apologize that due to tight regulation of radioactive materials and the outbreak of COVID-19 in China and the world, we cannot perform experiments the reviewer asked to direct test I846 and L843 in menthol binding. We hope the reviewer can understand our difficulties in this situation. We have removed the initial docking model to the closed state as the reviewer suggested below and in minor

comments #3. Docking of menthol to the open state is in agreement with our thermodynamic mutant cycle analysis as the distances between I846, L843 and menthol are all less than 4 Å (Fig. 1e in this revised version).

Regarding our thermodynamic mutant cycle analysis, we agree with the reviewer that this analysis cannot derive the unambiguous atomic model of menthol bound state, moreover, it is never our intention to state we have got such a model. Actually, what the thermodynamic mutant cycle analysis provided us are three pairs of protein-menthol interactions (R842-hydroxyl group of menthol, I846-isopropyl group of menthol and L843-isopropyl group of menthol). Based on these three pairs of interactions, we know how menthol molecule locally binds within its binding pocket. This is far away from any unambiguous atomic model of menthol bound state of the whole TRPM8 channel, and we are not claiming that we have got such a model from thermodynamic mutant cycle analysis. Our modeling efforts with ANAP imaging experiments have suggested a potential model of the open state TRPM8, but we totally agree with the reviewer that due to limitations in ANAP imaging, this model only represents one possible open state, but not the unambiguous atomic model of TRPM8 in the menthol bound open state. To avoid further confusion, as explained in our responses to the reviewer's previous concern, we have revised the text thoroughly and moved the Fig. 6 in the previous version, which shows our open state model, into the supplementary figures as Fig. S7.

3) Because the initial docking as well as MD simulation studies were still done using the model in the absence of the TRP domain. On the other side, they performed menthol docking using the open state model which is more energetically favored as they claimed (line 317-319). Therefore, the initial docking and MD simulation should be removed from the text, and the only the open state model should be described in the manuscript. In addition, the comparison of the menthol binding configuration between docking into apo state versus into the open-state model is over-interpretation and is not meaningful.

We agree with the reviewer and have removed the initial docking to apo state and MD simulation from the revised manuscript. In the revised manuscript, docking of menthol to the activated state is shown in Fig. 1e.

Minor comments

1) The authors tend to downplay the importance of the TRP domain in menthol- or ligand-dependent gating of TRPM8 as suggested by little change in their ANAP imaging results. Also, the authors' argument that the TRP domain is not critical for menthol binding is based on the previously publication by the LaTorre group (Brauchi, J. Neuroscience, 2006) as a supporting evidence. The TRP channel community is well aware of the issues with these studies (e.g. lack of reproducibility and suboptimal experimental design and interpretation) and has considered these studies not reliable. Throughout the manuscript revision, I can't help noticing that the authors tend to cite papers in a biased manner to support their claims (e.g. the previous citations that the authors used to support the selectivity filter gate theory, which authors has now removed upon my request). The functional importance of TRP domain is well supported by multiple solid studies by many groups (Patapoutian, Rohacs, Logothesis). Last, the position of the TRP domain differ upon different agonist binding in the reported structures.

We did not mean to downplay the importance of the TRP domain in the gating of TRPM8, we are fully aware that for ligands such as WS-12 and icilin to bind, the TRP domain is required as clearly shown in the cryo-EM structures. In fact, when we docked the smaller ligand menthol into its binding pocket, adding the TRP domain as previously suggested by the reviewer improved the docking score, which again shows the importance of the TRP domain. We just mean that in our ANAP imaging experiments, at the three functional ANAP-incorporated sites (Fig. 5f) we did not observe the significant shifts in ANAP emission peak, it is possible that conformational changes could be reported at other sites on the TRP domain, but these mutants with ANPA were non-functional. Indeed, we tested nine ANAP-incorporation mutants, but more than half of them (six out of nine) were non-functional (Supplementary Table 1). We have modified the text on these points (line 377-384). We have also added more literature citations regarding the roles of the TRP domain (line 383-384 and 578-580).

2) The Zagotta group has recently utilized ANAP coupled with TmFRET (Transition metal FRET) studies to obtain distance constraint as well as the directionality of motions associated with gating. With these information, they have successfully derived a model of HCN voltage gating (Dai et al, NSMB 2019). In the current studies by authors, fluorescence change of ANAP alone cannot generate much structural or dynamic information. The author should tone down the interpretation of the ANAP result and should address the limitation of their ANAP experiment.

We agree with the reviewer that TmFRET coupled with ANAP can generate distance information to directly constraint modeling building. In comparison, though ANAP alone cannot generate distance information as pointed out by the reviewer, it provides the SASA information which we used to filter out models that are inconsistent with ANAP information. We have explicitly stated this point (line 535-537, "SASA can be directly measured from a protein structure, so we can impose changes in SASA during computational modeling to filter out the models that are inconsistent with ANAP imaging results").

In this revision, as the reviewer suggested we have toned down our interpretation of ANAP constrained modeling and addressed the limitation of the ANAP experiments in discussion (line 306-309 and 405-411). As explained in response to reviewer's major concern #1, we have moved original Fig. 6 on the modeling results to the supplement figures. In the Results, we have changed the subtitle "Structural mechanisms underlying menthol activation" in the previous version to "Conformational dynamics in menthol activation" in this revised manuscript, where we limited our interpretation of the open state modeling to "Menthol binding may further lead to widening of the S6 bundle crossing" (line 324).

3) Line 229-236, the authors included statements to address the previous reviewer's comments regarding "L843 and I846 are outside interacting distance with menthol isopropyl group." Authors' reasoning in line 234-236 is incorrect. The distances (4.31 Å between L843, I846 and menthol) measured from the docking model do not reflect any conformational change. If the distances are over 4 Å in the ligand-bound model, it means residues L843 and I846 are unlikely to contribute significant binding with menthol. If the authors want to argue that the distance is averaged over a series of models from the simulation. Then what is the standard deviation of the distance measurement? The bottom line is that the difference between 4.31Å and below 4Å is in fact a big discrepancy.

The 4.31Å distance was measured from the initial docking model to the closed state. We agree with the reviewer's 3rd major concern that since docking of menthol to the activated state is energetically more

favorable, the initial docking can be confusing and unnecessary. We have removed the initial docking and MD simulation from the revised manuscript. In our docking to the activated state, the distances measured between menthol and L843 and I846 residues all agree with our thermodynamic mutant cycle analysis (Fig. 1e).

4) Lines 113-115. This statement is unnecessary. Multiple ligand-bound structures were published and showed agonists and antagonists binding to the cavity between S1-S4 and TRP domain. The authors based on sequence conservation and claimed more confident, which is redundant.

We totally agree with the reviewer that from published structures, it is very clear that ligands bind to the cavity between S1-S4 and TRP domain. However, as suggested by another reviewer, we performed sequence conservation analysis and the results are fully consistent with the structural studies. We did not mean “more confident”, in fact, we never used the word “more” in the text. To clarify on this point, we have modified this sentence (line 110-112).

5) I cannot find supplemental video as well as coordinates for the docked models in this revision. The only coordinates provided is for the newly built open state model.

As the reviewer suggested in the 3rd major concern, we removed docking of menthol to the closed state. Therefore, only the docking model of menthol to the open state is included in the supplementary files. Both the docking modeling to the closed state and the MD simulation video are removed.

Peer Review File

Reviewers' comments fourth round:

Reviewer #2 (Remarks to the Author):

In this revision, authors have improved the manuscript substantially by responding to most of my comments. I have the following minor comments, which authors should address before publication.

1) In my previous request (2) that authors should acknowledge the limit of using the thermodynamic mutant cycle analysis for modeling, which has led to the ambiguity of the menthol bound state in their model. However, although authors have fully agreed in their rebuttal letter, I found that they have not done this. Authors should state these points (technical limit and the model ambiguity) in their discussion, which I think would be important for this study to be viewed from the balanced perspective.

2) I requested to tone down the interpretation with ANAP studies. Although authors agreed and included a paragraph (405-411), but again they did not discuss the very reason for the limit of their ANAP studies for modeling studies: the lack of distance information. Instead, they described other factors (pH...) that may affect their accessibility measurement, which is not the point at all. Again, accessibility changes cannot be reliably integrated into atomic modeling. One cannot quantitatively convert accessibility change into distance changes or any meaningful changes in force fields. Therefore, authors should clearly address the technical limitation with their experimentally built open state model.

Reviewer #2 (Remarks to the Author):

In this revision, authors have improved the manuscript substantially by responding to most of my comments. I have the following minor comments, which authors should address before publication.

1) In my previous request (2) that authors should acknowledge the limit of using the thermodynamic mutant cycle analysis for modeling, which has led to the ambiguity of the menthol bound state in their model. However, although authors have fully agreed in their rebuttal letter, I found that they have not done this. Authors should state these points (technical limit and the model ambiguity) in their discussion, which I think would be important for this study to be viewed from the balanced perspective.

As the reviewer suggested, we have added discussion on the technical limit and the model ambiguity on thermodynamic mutant cycle analysis and ANAP imaging in the revised manuscript as we stated in the previous rebuttal letter (highlighted, line 420-428).

2) I requested to tone down the interpretation with ANAP studies. Although authors agreed and included a paragraph (405-411), but again they did not discuss the very reason for the limit of their ANAP studies for modeling studies: the lack of distance information. Instead, they described other factors (pH...) that may affect their accessibility measurement, which is not the point at all. Again, accessibility changes cannot be reliably integrated into atomic modeling. Once cannot quantitatively convert accessibility change in to distance changes or any meaningful changes in force fields. Therefore, authors should clearly address the technical limitation with their experiment.

We have explicitly stated that “shifts in ANAP emission spectrum cannot generate distance information between the fluorophore and residues of the channel protein to directly constrain structural modeling process” in Discussion (highlighted, line 414-416) and Methods (highlighted, line 558-561) of the revised manuscript.